# Acceleration-Dependent Effects of Vibrotactile Gamma Stimulation on Cognitive Recovery and Cholinergic Function in a Scopolamine-Induced Neurotoxicity Mouse Model

**DOI:** 10.3390/biomedicines13082031

**Published:** 2025-08-20

**Authors:** Tae-Woo Kim, Hee-Jung Park, Myeong-Hyun Nam, In-Ho Lee, Zu-Yu Chen, Hee-Deok Yun, Young-Kwon Seo

**Affiliations:** Department of Biomedical Engineering, Dongguk University, Goyang 10326, Republic of Korea; xodn8876@naver.com (T.-W.K.); gnflwldk98@naver.com (H.-J.P.); iis05047@naver.com (M.-H.N.); lih5537@naver.com (I.-H.L.); zuyuchen25@gmail.com (Z.-Y.C.); drengon1538@gmail.com (H.-D.Y.)

**Keywords:** vibrotactile stimulation, gamma oscillation, neurotoxicity syndromes, cognition disorders, cholinergic system, scopolamine, mice, synaptic plasticity, oxidative stress, signal transduction

## Abstract

**Background:** Alzheimer’s disease is a progressive neurodegenerative disorder characterized by cognitive decline and memory loss. Gamma (γ) oscillations are closely linked to learning and memory, and recent interest has grown around Gamma ENtrainment Using Sensory stimulation (GENUS) as a non-invasive neuromodulation strategy. However, the therapeutic impact of vibrotactile gamma stimulation under varying physical parameters such as acceleration remains underexplored. **Methods:** Differentiated SH-SY5Y cells were treated with amyloid-β (Aβ) and exposed to vibrotactile stimulation at 2.2 or 4.0 m/s^2^. In vivo, male C57BL/6N mice (7 weeks old, 35 g) were administered scopolamine to induce neurotoxicity and randomly assigned to sham, scopolamine, donepezil, or vibrotactile stimulation groups (n = 10 each). Behavioral tests, biochemical assays, Western blotting, and immunohistochemistry were performed to evaluate cognitive function, oxidative stress, cholinergic activity, synaptic plasticity, and neuroinflammation. **Results:** In vitro, SH-SY5Y cells exposed to amyloid-beta (Aβ) were treated with vibrotactile stimulation, resulting in enhanced neuronal marker expression at 2.2 m/s^2^. In vivo, mice receiving stimulation at 2.2 m/s^2^ showed improved cognitive performance, reduced oxidative stress, restored cholinergic function, suppressed neuroinflammation, and enhanced synaptic plasticity. Mechanistically, these effects were associated with activation of the AKT/GSK3β/β-catenin pathway. **Conclusions:** Our findings demonstrate that vibrotactile gamma stimulation at 2.2 m/s^2^ exerts greater therapeutic efficacy than higher acceleration, highlighting the importance of optimizing stimulation parameters. This work supports the potential of acceleration-tuned, non-invasive GENUS-based therapies as effective strategies for cognitive recovery in neurodegenerative conditions.

## 1. Introduction

Alzheimer’s disease is a prevalent neurodegenerative disorder characterized by progressive memory loss and cognitive decline [1,2]. While the exact etiology of AD remains unclear, its pathology is associated with several key factors, including the aggregation of amyloid-beta (Aβ) leading to the formation of senile plaques. This process contributes to the degeneration and loss of cholinergic neurons, resulting in decreased acetylcholine (ACh) levels, a hallmark of dementia [3,4]. Additionally, oxidative stress, inflammatory responses, and neuronal death play crucial roles in the progression of AD, ultimately leading to cognitive impairment [5,6].

Scopolamine, a nonselective antimuscarinic agent, functions as a competitive antagonist of muscarinic acetylcholine receptors [7,8]. By disrupting central cholinergic neurotransmission, scopolamine significantly impairs learning acquisition and short-term memory. Moreover, it induces oxidative stress by generating reactive oxygen species (ROS), contributing to neuronal damage [9,10]. Studies have also demonstrated that scopolamine promotes neuronal apoptosis in the hippocampus, a brain region essential for learning and memory [11,12,13].

The AKT/GSK-3β/β-catenin pathway is a key regulator of neuronal survival and synaptic plasticity. Activation of AKT suppresses pro-apoptotic signaling and inhibits GSK-3β by phosphorylation at Ser9 [14,15,16]. In contrast, aberrant GSK-3β activity promotes cholinergic deficits, Aβ accumulation, and tau hyperphosphorylation [17,18]. When GSK-3β is inhibited, β-catenin becomes stabilized, activating Wnt/β-catenin signaling that supports neuronal regeneration and reduces Aβ aggregation [19,20]. Dysregulation of this cascade is therefore closely linked to Aβ-related neurotoxicity and cognitive decline.

Gamma (γ) oscillations are closely associated with learning and memory processes. Gamma oscillations strengthen synaptic plasticity and promote efficient information transfer across brain regions. They also organize the temporal structure of memory traces, supporting both memory formation and retrieval. Furthermore, by enhancing synchronization between the prefrontal cortex and hippocampus, gamma activity contributes to higher-order cognitive functions such as learning, attention, and working memory [21,22]. Notably, studies have demonstrated that AD patients exhibit a reduction in gamma oscillation power, a phenomenon that has also been replicated in various AD mouse models [23,24,25]. Given these findings, recent research has focused on Gamma ENtrainment Using Sensory stimulation (GENUS), a non-invasive sensory stimulation approach aimed at enhancing gamma oscillations as a potential therapeutic strategy for AD and other neurodegenerative disorders.

GENUS utilizes various forms of non-invasive stimuli to induce gamma oscillations. Electromagnetic stimulation has been shown to enhance cholinergic function in patients with AD [23] and to alleviate abnormal connectivity between the hippocampus and prefrontal cortex while increasing gamma oscillations in APP/PS1 mice [24]. Auditory stimulation has been reported to reduce Aβ deposition, inhibit GSK-3β, restore mitochondrial function, and improve cognitive performance in APP/PS1 and 5xFAD mouse models [25,26]. Similarly, photostimulation decreases amyloid-beta accumulation, attenuates neuronal and synaptic loss, and exerts neuroprotective effects in mouse models [27,28].

Recently, studies have explored vibrotactile stimulation as a means of modulating gamma activity. A previous study showed that six weeks of 40 Hz whole-body vibration at ~1.01 m/s^2^ improved motor performance and reduced synaptic loss and DNA damage in tau P301S and CK-p25 mice [29]. Another study demonstrated that eight weeks of whole-body vibration at 40 Hz with a vertical amplitude of 0.8 mm in aged male rats enhanced cognitive function and reduced neuronal apoptosis [22]. Among various stimulation frequencies, 40 Hz has uniquely been shown to induce gamma oscillations and synchronize neural activity, which are crucial for cognitive function [22,28,29]. Notably, 40 Hz also falls within the sensitive range of Piezo1, a mechanosensitive ion channel that responds to mechanical vibrations between 1 and 100 Hz [30]. Activation of Piezo1 can trigger intracellular signaling pathways that regulate oxidative stress, cell death, and immune responses in microglia [31]. Since parameters such as stimulation duration, frequency, and acceleration determine the activation of sensory receptors such as Piezo1 and influence systemic physiological responses, it is reasonable to expect that differences in acceleration may lead to distinct neurobiological outcomes.

Although GENUS has shown promising neuroprotective effects, the precise mechanisms by which it influences amyloid-β deposition remain incompletely understood. Some studies have reported that gamma entrainment or GENUS promotes a phagocytic phenotype in microglia, thereby enhancing their clearance activity and alleviating neuroinflammation, while others have demonstrated that 40 Hz stimulation reduces amyloid burden through a combination of decreased APP cleavage intermediates in hippocampal neurons, increased microglial endocytosis, and suppression of inflammatory responses [26,27,28]. However, the cellular and molecular processes by which such gamma synchronization leads to memory improvement remain largely unresolved and represent an important topic for future investigation.

A recent clinical study reported that continuous vibrotactile stimulation at 15–40 Hz for one month led to significant improvements in memory, executive function, and processing speed in individuals with mild cognitive impairment or mild dementia [32]. Another study provided compelling neurophysiological evidence for the modulation of gamma oscillations through a single session of TMS in patients over 65 years of age [33]. These findings highlight that sensory entrainment therapies are increasingly being explored in clinical settings and are emerging as promising approaches for novel therapeutic interventions. Recent developments in wearable neuromodulation technologies support the potential clinical translation of vibrotactile stimulation. For instance, the PanBrain EC2 (PanBrain Tech Co., Ltd., Singapore) is a wearable headband-type tDCS device designed for cognitive and emotional regulation, highlighting the feasibility of developing non-invasive tools. Additionally, several clinical studies have explored the cognitive benefits of whole-body vibration (WBV) exercisers [34,35], further supporting the applicability of such devices in therapeutic interventions. These examples suggest that vibrotactile stimulation could be adapted into wearable or platform-based systems for human cognitive rehabilitation.

Nevertheless, several clinical limitations remain. Gamma rhythms can be induced through various approaches, and the outcomes as well as the underlying mechanisms may differ depending on the induction method. For instance, 40 Hz photostimulation in mouse models has been reported to attenuate AD pathology [27,36], but in other cases, it has been associated with increased amyloid deposition in mice [37]. Such evidence indicates that even with the same type of stimulation, variations in stimulation parameters may differentially influence disease pathology, potentially eliciting diverse molecular and cellular responses [38]. Therefore, for clinical translation, it is essential to determine which stimulation strategies are beneficial for specific patient groups, and future studies should systematically examine the mechanisms and therapeutic effects of gamma entrainment across a wide range of stimulation parameters. In studies of vibrotactile stimulation, there are also reports that gamma-frequency stimulation shows no effect or only partial effects in alleviating Alzheimer’s disease symptoms [39,40,41,42]. These inconsistent results appear to be largely due to the absence of standardized stimulation protocols. In fact, whole-body vibration (WBV), a form of vibrotactile stimulation, varies greatly across studies in terms of vibration frequency, acceleration, amplitude, and duration [43].

To address this issue, the present study aimed to investigate the importance of acceleration at a fixed frequency, as well as how different acceleration levels affect therapeutic outcomes. We employed a scopolamine-induced neurotoxicity mouse model and an Aβ oligomer-treated SH-SY5Y cell system, which mimic key aspects of Alzheimer-like pathology such as cholinergic dysfunction and Aβ-related toxicity. We examined the recovery of Aβ-induced cytotoxicity under varying accelerations using RT-qPCR in vitro. We further investigated the acceleration-dependent effects of vibrotactile stimulation on cognitive functions, as well as associated molecular changes in the brain, in a scopolamine-induced neurotoxicity mouse model. Behavioral and molecular analyses revealed that vibrotactile stimulation improved cognitive function and reversed scopolamine-induced neuropathological changes, including apoptosis, neuroinflammation, cholinergic dysfunction, and impaired synaptic plasticity, via acceleration-dependent activation of neuroprotective signaling pathways.

## 2. Materials and Methods

This study employed both in vitro and in vivo approaches to evaluate the acceleration-dependent therapeutic effects of 40 Hz vibrotactile stimulation on cognitive and molecular outcomes in a neurotoxicity model. In vitro, differentiated SH-SY5Y cells were treated with amyloid-beta (Aβ) and exposed to vibrotactile stimulation at different acceleration levels. In vivo, male C57BL6/N mice received chronic scopolamine injections to induce neurotoxicity, followed by vibrotactile stimulation or donepezil treatment. Behavioral assessments, biochemical assays, and histological analyses were conducted to investigate cognitive function, cholinergic activity, oxidative stress, synaptic plasticity, and neuroinflammation. This comprehensive design allowed for the evaluation of both behavioral and mechanistic effects of stimulation across biological levels.

### 2.1. Cell Culture and Treatment

The human neuroblastoma SH-SY5Y cell line was cultured in high-glucose DMEM supplemented with 10% fetal bovine serum (FBS) and 1% penicillin–streptomycin (PS) at 37 °C in a humidified atmosphere with 5% CO_2_. For neuronal differentiation, 1 × 10^6^ cells were seeded in 100 mm dishes and cultured for 6 days in differentiation media consisting of DMEM/F12 with 1% FBS, 1% PS, and 10 μM retinoic acid. After differentiation, cells were divided into four groups: (1) a control group without Aβ treatment (Aβ-), (2) an Aβ-treated group (5 μM; Aβ+), and (3–4) two groups treated with 5 μM Aβ oligomers and subjected to 40 Hz vibrotactile stimulation at either 2.2 m/s^2^ (335 mV) or 4 m/s^2^ (570 mV) for 30 min daily over 3 days. Aβ1-42 “click peptide” (GenScript) was used to produce Aβ oligomers. For oligomerization, the peptide was diluted to 5 μM in PBS and incubated at 37 °C for 3 h on a 300 RPM shaker. The treatment for each experimental group was conducted for three days after differentiation. All treatments were conducted under consistent culture conditions. To quantitatively assess dendritic morphology, neurite length was measured from phase-contrast images using ImageJ (version 1.54k).

### 2.2. Reverse Transcription Followed by Quantitative PCR

Total RNA was extracted from cells 30 min after the final day of vibrotactile stimulation over a 3-day period, using RNAiso Plus (1 mL per sample; Takara Bio, Shiga, Japan). Chloroform (200 μL; Sigma, Tokyo, Japan) was added to the lysate, followed by incubation at room temperature for 3 min. The mixture was centrifuged at 12,000× *g* at 4 °C for 15 min, and the aqueous phase was carefully collected. Isopropanol (500 μL) was added to the collected supernatant and incubated at room temperature for 10 min. The mixture was centrifuged at 12,000× *g* for 10 min, and the resulting pellet was washed with 1 mL of 75% ethanol. After centrifugation at 7500× *g* at 4 °C for 5 min, the supernatant was removed, and the pellet was air-dried at room temperature. The dried RNA pellet was dissolved in 20 μL of RNase-free water and heated at 65 °C for 10 min using a heat block. The RNA concentration was determined using a NanoDrop spectrophotometer (Thermo Fisher Scientific, Waltham, MA, USA). cDNA was synthesized from the extracted RNA using a reverse transcription master mix (Dynebio, Seattle, WA, USA). Quantitative PCR (qPCR) was performed using the StepOnePlus™ real-time PCR system (Applied Biosystems, Thermo Fisher Scientific, Waltham, MA, USA) with TB Green^®^ Premix Ex Taq™ (Takara Bio, Shiga, Japan). Gene expression levels were analyzed using the comparative CT (ΔΔCT) method. RT-qPCR was performed using the following primer sets: *MAP2*—Forward: 5′-TCA GAG GCA ATG ACC TTA CC-3′; Reverse: 5′-GTG GTA GGC TCT TGG TCT TT-3′; *NeuroD1*—Forward: 5′-GGT GCC TTG CTA TTC TAA GAC GC-3′; Reverse: 5′-GCA AAG CGT CTG AAC GAA GGA G-3′; *PSD95*—Forward: 5′-CGG GCG GGA TTA AGG AGT TT-3′; Reverse: 5′-AAC CCT GAC TCA TCG TCC AC-3′. Each experimental condition was analyzed in three independent biological replicates, and within each biological replicate, three technical replicates were performed. For data analysis, the mean value of the three technical replicates was calculated for each biological replicate. These averaged values were then used to compute the mean ± standard error of the mean (SEM) across the three biological replicates, and individual data points in the figures represent each biological replicate’s averaged value.

### 2.3. Animals

Male C57BL/6N mice (aged 7 weeks, weighing 35 g) were obtained from Dae Han Biolink Co., Ltd. (Eumseong, Republic of Korea). They were maintained under controlled temperature (22 °C ± 2 °C) with a 12 h shifting light–dark cycle. The mice were housed five per cage with free access to food and water and provided with environmental enrichment such as nesting materials. Animals were monitored daily for general health, body weight, and behavior. Humane endpoints were applied based on predefined clinical scoring criteria (Appendix A), and all animals used in this study did not exhibit any conditions warranting euthanasia. All animal procedures were approved by the Institutional Animal Care and Use Committee of Dongguk University (IACUC-2023-024-2, approval date: 11 September 2023).

### 2.4. Experimental Design

Mice were randomly grouped into five groups: (1) sham group (Sham, n = 10); (2) scopolamine-treated group (Scopolamine, n = 10); (3) scopolamine- and donepezil-treated group (Donepezil, n = 10): (4) scopolamine and vibrotactile stimulation acceleration 2.2 m/s^2^-treated group (2.2 m/s^2^, n = 10); (5) scopolamine and vibrotactile stimulation acceleration 4 m/s^2^-treated group (4 m/s^2^, n = 10). Group sizes (n = 10) were preliminarily set based on consideration of statistical robustness, with reference to previous studies [44,45,46]. Although no formal power analysis was conducted, a minimum of eight animals per group was deemed sufficient, and ten animals per group were included to account for potential variability and dropout. Before the experiment, a preliminary motor performance test was conducted, and mice with significantly reduced motor ability were excluded. Based on the results, the selected mice were randomly assigned to experimental groups. All animals were numerically coded, and the researchers performing behavioral tests and tissue analyses were blinded to group assignments. We used only male mice in this study to minimize the potential confounding effects of hormonal fluctuations associated with the estrous cycle, which can influence behavioral and molecular outcomes. However, we acknowledge that this design limits the generalizability of our findings to both sexes. The sham group received an intraperitoneal injection of saline (3 mg/kg body weight) and were placed on the vibration platform without stimulation for 30 min per day. Scopolamine hydrobromide was dissolved in saline at a concentration of 0.3 mg/mL and administered intraperitoneally at a volume of 10 µL/g. Donepezil (10 mg) was dissolved in 100 µL DMSO and diluted with saline to 33 mL, resulting in a 0.3 mg/mL solution with 0.3% DMSO (*v*/*v*). Donepezil was administered orally at 10 µL/g, 15 min after scopolamine injection. The oral route of administration was selected based on previous studies using mouse models [44,47,48]. Additionally, we aimed to minimize potential interactions with scopolamine, which was administered via intraperitoneal injection. All drugs were administered at a final dose of 3 mg/kg body weight. All groups except the sham group received daily scopolamine injections. The vibrotactile stimulation was also applied 30 min after scopolamine injection and continued for 30 min. Scopolamine administration was conducted for 9 weeks, and vibrotactile stimulation and donepezil treatment were applied starting two weeks after the initiation of scopolamine administration, followed by behavioral test. All behavioral tests were conducted 1 h after scopolamine injection, with donepezil administered 15 min after scopolamine injection, and vibrotactile stimulation applied 30 min after scopolamine injection (Figure 1A). Mice were sacrificed immediately after the completion of the behavioral tests for subsequent biochemical analyses, Western blotting, and immunohistochemistry.

### 2.5. Vibrotactile Stimulation

A sinusoidal waveform with a frequency of 40 Hz was supplied to a circular-shaped vibrator with a maximum output of 5 W from the function generator (EZ Digital, FC-7002C, Anyang, Republic of Korea). The circular vibration plate, with a diameter of 12 cm, was designed to deliver vibrations to either a 100 pi dish or the entire body of a mouse placed on it. When applying vibrations to cells, the dish was secured using rubber bands attached to the screws on both sides for stable stimulation (Figure 1A). For mice, a cylindrical plastic barrier was set up around the vibration plate to prevent them from leaving the area (Figure 1B). A single vibrator could accommodate up to four mice at a time. Vibrotactile stimulation was applied daily for 30 min, with accelerations of either 2.2 m/s^2^ or 4 m/s^2^, depending on the experimental group. Each acceleration value was measured using an accelerometer (Dytran Instruments, Inc., 3225F, Chatsworth, CA, USA) with a sensitivity of 10 mV/g, a measurement range up to 500 g, and a frequency response of 1.6–10,000 Hz (±10%) prior to applying vibrotactile stimulation.

### 2.6. Behavior Test

#### 2.6.1. Morris Water Maze

The Morris water maze (MWM) test was performed in a circular open pool filled with water, with the temperature maintained at 22 °C ± 1 °C. The swimming time, path record, and data analysis were conducted using the ANY-maze video tracking system. A platform was positioned at the center of one of the quadrants, which was divided into four equal sections by two primary vertical axes. The mice started in the diagonal quadrant where the platform was located, and the experiment was conducted from there. The training sessions were conducted for 3 days, and a test session was performed on day 4. During the training sessions, the mice swam for 60 s to find the platform. If they were unable to locate the platform within the time limit, they were given a 30 s acquisition period by placing them on the platform. The test session followed the same procedure, except that no acquisition period was provided if the mice were unable to find the platform. Before the start of drug administration and behavioral experiments, all mice were pre-screened for swimming ability and general motor performance. Based on these assessments, animals were randomly assigned to experimental groups to ensure balanced baseline motor function across groups, and we carefully standardized all experimental conditions to minimize stress-related variability across groups. During the MWM training, all mice were monitored for swimming ability and motor coordination, and no group differences were observed. Moreover, previous studies have also employed MWM following vibrotactile stimulation and reported valid cognitive outcomes, further supporting its suitability in this experimental context [22].

#### 2.6.2. Novel Object Recognition Test

The novel object recognition (NOR) test was conducted in a square chamber made of polyvinyl plastic with each side measuring 25 cm (25 cm × 25 cm × 25 cm). During the habituation, the mice were placed in the chamber without any objects for 5 min. During the test, two identical objects were placed in the chamber, and the mice investigated them for 5 min. Four hours later, one of the two identical objects was replaced with a novel object, and the mice were reintroduced into the chamber for 5 min. The ANY-maze video tracking system recorded the time spent by the animals exploring each object. (Time spent exploring the novel object/total exploration time for both objects) × 100 as the formula used to calculate the percentage of recognition index (%).

#### 2.6.3. Y-Maze Test

The Y-maze test was conducted in a maze with three arms, made of polyvinyl plastic. The mouse was placed at the center of the maze and allowed to explore freely for 7 min. The entry of the mouse’s hind legs into the arms of the maze was recorded as an entry into that arm. (The number of consecutive entries into the three different arms/the total number of arm entries − 2) × 100 was the formula used to calculate the percentage of alternation score (%).

### 2.7. Lipid Peroxidation

Lipid peroxidation, an indicator of oxidative stress, was measured using the Peroxidation (TBARS) Assay Kit (DoGenBio, Seoul, Republic of Korea). The cerebral cortex was isolated and homogenized in PBS containing heparin. Malondialdehyde (MDA), a key marker of lipid peroxidation, reacts with thiobarbituric acid (TBA) to form MDA–TBA adducts. The optical density (OD) of each sample and standard was measured at 540 nm, and the OD of the blank was subtracted. Duplicate readings were averaged, and MDA concentrations were calculated using a standard curve generated from known MDA concentrations. The standard curve was generated by performing a 2-fold serial dilution starting from 400 μM, and the resulting graph is presented in the Appendix A. Final values were adjusted for any dilution factor and normalized to total protein concentration, expressed as μM/mg protein.

### 2.8. AChE Activity

Acetylcholinesterase (AChE) is an enzyme that hydrolyzes the neurotransmitter ACh into choline and acetate. Acetylcholinesterase (AChE) activity was evaluated using the Amplite^®^ Colorimetric Acetylcholinesterase Assay Kit (AAT Bioquest, Inc., Santa Clara, CA, USA). According to the manufacturer’s instructions, cortical tissue was homogenized in PBS, and 50 µL of sample or standard was mixed with 50 µL of working solution containing DTNB and acetylthiocholine. After incubation at room temperature for 30 min, absorbance was measured at 410 nm. A standard curve was generated by performing 1:3 serial dilutions starting from a 1000 mU/mL acetylcholinesterase standard solution. The resulting values were analyzed using an online linear regression calculator, and the curve was plotted on a semi-logarithmic scale (Appendix A). The activity in the samples was calculated by interpolation using the linear regression equation derived from the standard curve.

### 2.9. Western Blotting

Mouse hippocampi were isolated bilaterally and homogenized in sample buffer containing 0.25 M Tris-HCl (pH 6.8), 4% SDS, 40% glycerol, 0.05% bromophenol blue, and 10% β-mercaptoethanol. The homogenates were centrifuged at 12,000× *g* for 15 min at 4 °C, and the supernatants were collected. Protein concentrations were measured using a BCA assay by generating a standard curve with bovine serum albumin (BSA) and calculating the protein concentration of samples based on their absorbance values. For SDS-PAGE, the sample was loaded onto a 10% SDS-polyacrylamide gel and electrophoresed at 90 V for 120 min. The average protein concentration was approximately 6 µg/µL. To load 30 µg of protein per lane, 4.9 µL of lysate was mixed with loading buffer to a final volume of 20 µL per well. All Western blot experiments were performed using identical protein extracts from the same set of samples, which were loaded in equal amounts for each target protein on separate gels. The proteins were transferred onto a polyvinylidene difluoride (PVDF) membrane. The membrane was blocked in 5% skim milk for 1 h at room temperature and washed three times with TBS-T for 10 min each. The membrane was then incubated for 1 h with primary antibody. After washing with TBS-T, the blots were incubated with a secondary antibody in 5% skim milk for 2 h at room temperature. Protein bands were detected using enhanced chemiluminescence (ECL) and captured with a ChemiDoc XRS+ imaging system (Bio-Rad, Hercules, CA, USA). The primary and secondary antibodies used for Western blotting, including their dilution ratios, sources, and catalog numbers, are listed in Appendix A. A single β-actin blot obtained from one of these gels was used as a representative loading control for all target proteins because the loading conditions and samples were identical across all experiments.

### 2.10. Immunohistochemical Analyses

Mouse brain tissues were fixed in 4% paraformaldehyde (PFA) overnight at 4 °C. The fixed tissues were dehydrated through a series of graded alcohol solutions and embedded in paraffin. Paraffin-embedded tissues were sectioned at a thickness of 4 μm, mounted onto slides, and dried at room temperature. The slides were incubated at 65 °C for 2 h. After deparaffinization and rehydration, the slides were incubated with hydrogen peroxide at room temperature for 10 min, followed by rinsing with PBS-T (phosphate-buffered saline with 0.1% Tween-20). The sections were then incubated with primary antibodies at 4 °C overnight. Following primary antibody incubation, the slides were exposed to anti-rabbit/mouse HRP-conjugated secondary antibodies (Agilent, #K5007, Santa Clara, CA, USA) at room temperature for 30 min. Color development was performed using DAB (Agilent Dako, #K5007) for 3 min for all markers, except for BDNF and PSD95, which required 4 min to achieve optimal signal. The sections were rinsed with PBS-T and counterstained with Mayer’s Hematoxylin (MUTO PURE CHEMICALS, #30002, Tokyo, Japan) for 4 min. The images were captured using a light microscope. The primary and secondary antibodies used for immunohistochemistry, including their dilution ratios, sources, and catalog numbers, are listed in Appendix A.

### 2.11. Statistical Analysis

All experimental data are presented as mean ± standard error. Differences between groups were analyzed using a one-way analysis of variance (ANOVA) followed by Tukey’s multiple comparisons test. A *p*-value of less than 0.05 was considered statistically significant. *p*-values of less than 0.05, 0.01, 0.005, and 0.001 were indicated by *, **, ***, and ****, respectively.

## 3. Results

### 3.1. Effects of Vibrotactile Stimulation on Aβ-Induced Neural Damage In Vitro

Preliminary in vitro screening using SH-SY5Y cells demonstrated that 40 Hz vibrotactile stimulation at 2.2 m/s^2^ produced the most robust upregulation of neuronal differentiation markers, including *GAP-43*, *DCX*, *NeuroD1*, and *MAP2*, compared to other tested accelerations (1.6 and 5.9 m/s^2^) (Appendix A). Based on these findings, 2.2 m/s^2^ was selected as the primary condition for in vivo experiments. Additionally, 4.0 m/s^2^—intermediate between 2.2 and 5.9 m/s^2^—was chosen as a secondary condition to investigate potential nonlinear effects of stimulation intensity

To investigate the effects of vibrotactile stimulation on neural damage, SH-SY5Y cells were differentiated into neuronal-like cells and exposed to Aβ to induce neurotoxicity. As shown in Figure 2A,B, the Aβ+ group exhibited a marked reduction in dendritic structures compared to the untreated Aβ− control group. However, both the 2.2 m/s^2^ (V2.2) and 4 m/s^2^ (V4) vibrotactile stimulation groups demonstrated significant restoration of dendritic structures (Figure 2C–E). To further evaluate neuronal recovery, the expression levels of neuronal markers *MAP2*, *NeuroD1*, and *PSD95* were quantified using RT-qPCR (Figure 2F–H*). MAP2* and *NeuroD1* expression levels were reduced in the Aβ+ group compared to the Aβ− group but were restored in the V2.2 group. *PSD95* expression was markedly increased in the V2.2 group compared to the Aβ+. However, no significant changes were observed in the V4 group.

### 3.2. Effects of Vibrotactile Stimulation on Cognitive Function in a Scopolamine-Induced Alzheimer’s Disease Model

To investigate whether vibrotactile stimulation can improve cognitive function in a scopolamine-induced neurotoxicity model, we conducted the MWN, NOR test, and Y-maze test. In the MWM, the reduction in escape latency over four days was the smallest in the scopolamine-treated group, indicating impaired spatial learning. However, the V2.2 and V4 vibrotactile stimulation groups exhibited a significant reduction in escape latency compared to the scopolamine-treated group, with 1.84-fold and 1.88-fold decreases, respectively (Figure 3A–C). Analysis of swimming paths and escape latency on the test day further revealed that scopolamine treatment resulted in more complex swimming trajectories and a markedly increased escape latency compared to the sham group. In contrast, the donepezil-treated and vibrotactile stimulation groups exhibited swimming patterns and escape latency comparable to those of the normal mice. In the NOR test, the recognition index was significantly higher in the V2.2 group than in the scopolamine group, showing a 2.01-fold increase, suggesting an improvement in recognition memory (Figure 3D). No statistically significant differences in alternation scores were observed in the Y-maze test. (Figure 3E). Detailed statistical results are provided in Appendix A.

### 3.3. Biochemical Analysis of Oxidative Stress and Cholinergic Function in the Brain Following Scopolamine and Vibrotactile Stimulation

To investigate the biochemical changes in the brain induced by scopolamine and the effects of vibrotactile stimulation in vivo, we conducted biochemical assays targeting oxidative stress and cholinergic function in the cortex. To assess oxidative stress, a lipid peroxidation assay was performed to measure MDA levels in the cortex. The cortical tissues from multiple animals in each group were pooled and reduced to two data points, which were presented as average values. The MDA level was significantly increased in the scopolamine-treated group, indicating elevated oxidative stress. However, vibrotactile stimulation at V2.2 significantly reduced MDA levels by 2.1-fold compared to the scopolamine group (*p* < 0.0001, Figure 4A).

To evaluate changes in cholinergic function, an AChE assay was conducted in the cortex. AChE activity was markedly elevated in the scopolamine group compared to the sham group, suggesting impaired cholinergic function. However, vibrotactile stimulation at V2.2 and V4 restored AChE activity to levels comparable to the normal group, showing 1.3-fold reductions compared to the scopolamine group (*p* = 0.0084 for V2.2, *p* = 0.0109 for V4) (Figure 4B).

### 3.4. Immunohistochemical Analysis of Aβ, BAX, and AChE Expression in the Hippocampus of a Scopolamine-Induced AD Mouse Model Following Vibrotactile Stimulation

To assess the expression of Aβ, BAX, and AChE in the hippocampus of a scopolamine-induced neurotoxicity mouse model, immunohistochemical staining was performed on brain sections (Figure 5A–O).

A key pathological hallmark of AD, amyloid-β (Aβ) plaques, was markedly increased in the scopolamine-treated group (Figure 5B, arrow). However, treatment with donepezil or vibrotactile stimulation (V2.2, V4) significantly reduced Aβ plaque formation (Figure 5C–E). Similarly, the expression of BAX, a pro-apoptotic marker, was elevated in the scopolamine group, indicating increased neuronal apoptosis. In contrast, BAX expression was reduced in the donepezil and vibrotactile stimulation (V2.2, V4) groups, suggesting a neuroprotective effect (Figure 5F–G). Furthermore, scopolamine-induced upregulation of AChE expression was restored to levels comparable to the sham group following treatment with donepezil or vibrotactile stimulation (Figure 5H–K).

### 3.5. Immunohistochemical Analysis of Neuroinflammation in a Scopolamine-Induced Alzheimer’s Disease Mouse Model Following Vibrotactile Stimulation

To evaluate the effects of vibrotactile stimulation on neuroinflammation, the expression levels of inflammatory markers IL-1β and TNF-α, and the microglial marker IBA-1 were analyzed in the hippocampus using immunohistochemistry (Figure 6A–O). Both IL-1β and TNF-α expression were significantly elevated in the scopolamine-treated group compared to the sham group, indicating increased neuroinflammation (Figure 6A,B,F,G). However, this increase was markedly reduced in the donepezil-treated and vibrotactile stimulation (V2.2, V4) groups, restoring expression levels to those observed in the sham group (Figure 6C–E,H–J). Additionally, microglial activation, a key driver of neuroinflammation, was assessed by IBA-1 staining. The scopolamine-treated group exhibited a notable increase in microglial proliferation (Figure 6L), whereas vibrotactile stimulation (V2.2, V4) significantly reduced microglial activation to levels comparable to the sham group, similar to the effects observed in the donepezil-treated group (Figure 6K,M–O). These findings suggest that vibrotactile stimulation effectively alleviates scopolamine-induced neuroinflammation, demonstrating a potential therapeutic effect in reducing neuroinflammatory responses in neurotoxicity.

### 3.6. Immunohistochemical Analysis of Neural Recovery in a Scopolamine-Induced Alzheimer’s Disease Mouse Model Following Vibrotactile Stimulation

The neuroprotective effect of vibrotactile stimulation was investigated through immunohistochemical analysis of BDNF and PSD95 expression in the hippocampus in a scopolamine-induced AD mouse model (Figure 7A–F,J). In the scopolamine group, the expression of BDNF was almost undetectable and significantly lower compared to the sham group (Figure 7A,B). However, vibrotactile stimulation notably increased BDNF expression, with the 2.2 m/s^2^ vibration intensity showing a level similar to that of the sham group (Figure 7D,E). PSD95 expression also showed the highest increase at 2.2 m/s^2^, significantly higher than in the scopolamine group (Figure 7I,G).

### 3.7. Effects of Vibrotactile Stimulation on Neuroinflammation and Apoptosis-Related Proteins in the Hippocampus of a Scopolamine-Induced AD Mouse Model

To more precisely investigate the effects of vibrotactile stimulation on neuroinflammation and apoptosis, we analyzed the protein levels of IL-1β and BAX in the hippocampus using Western blotting (Figure 8A–C). The expression of IL-1β was significantly increased in the scopolamine group compared to the sham group, showing a 4.18-fold increase (*p* < 0.0001). However, both the donepezil and vibrotactile stimulation (V2.2, V4) groups exhibited a significant reduction in IL-1β levels (Figure 8B). Similarly, BAX was markedly upregulated in the scopolamine group compared to the sham group, showing a 2.84-fold increase, indicating enhanced neuronal apoptosis (*p* = 0.0002). Treatment with donepezil or vibrotactile stimulation significantly mitigated this increase, restoring BAX expression to levels comparable to the sham group. Notably, the V2.2 group demonstrated a greater reduction in BAX expression than the other treatment groups, with a 2.98-fold decrease compared to the scopolamine group (*p* = 0.0001) (Figure 8C).

### 3.8. Effects of Vibrotactile Stimulation on Cholinergic Function in the Hippocampus of a Scopolamine-Induced AD Mouse Model

To assess the impact of vibrotactile stimulation on cholinergic function, we analyzed the expression levels of GAP43, PSD95, NGF, synaptophysin (SYN), and choline acetyltransferase (ChAT) in the hippocampus using Western blotting (Figure 9A–F).

The expression of GAP43 and PSD95 was significantly increased in the donepezil group compared to the scopolamine group. Among the vibrotactile stimulation groups, the acceleration of 2.2 m/s^2^ showed a comparable increase to the donepezil group, with 1.44-fold and 1.58-fold increases in GAP43 and PSD95 expression, respectively (Figure 8B,C). Notably, both GAP43 and PSD95 expression levels were significantly higher in the V2.2 group compared to the V4 group (*p* = 0.0298 and *p* < 0.0001, Appendix A). Similarly, NGF expression, which is essential for neuronal survival and function, was significantly decreased in the scopolamine group. However, treatment with donepezil or vibrotactile stimulation effectively restored NGF levels. Specifically, NGF expression increased by 1.91-fold in the V2.2 group and 2.50-fold in the V4 group compared to the scopolamine group (Figure 9D). SYN, a presynaptic marker indicative of synaptic integrity, was also markedly reduced in the scopolamine group. Both donepezil and vibrotactile stimulation significantly increased SYN expression, with a 4.87-fold increase in the V2.2 group and a 6.40-fold increase in the V4 group relative to the scopolamine group (Figure 9E).

Lastly, ChAT, a key enzyme in cholinergic neurotransmission, was markedly downregulated in the scopolamine group. Both donepezil and vibrotactile stimulation restored ChAT expression. Notably, the V2.2 group exhibited a significantly greater increase in ChAT levels compared to the V4 group, with a 4.89-fold versus a 3.07-fold increase, respectively (Figure 9F). This difference was statistically significant (*p* = 0.0044), indicating the superior efficacy of the 2.2 m/s^2^ stimulation (Appendix A).

### 3.9. Effects of Vibrotactile Stimulation on the AKT/GSK3β/β-Catenin Pathway in the Hippocampus of a Scopolamine-Induced AD Mouse Model

To further investigate the mechanism of vibrotactile stimulation, we analyzed the expression levels of proteins involved in the AKT/GSK3β/β-catenin signaling pathway in the hippocampus using Western blotting (Figure 10A–D). The expression of p-AKT was significantly increased in the V2.2 and V4 stimulation groups compared to the scopolamine group, showing 2.08-fold and 3.81-fold increases, respectively (Figure 10B). Phosphorylated GSK3β (p-GSK3β) levels, which were significantly reduced in the scopolamine group, were restored by vibrotactile stimulation. The V2.2 and V4 groups showed 1.89-fold and 1.86-fold increases, respectively (Figure 10C). β-Catenin expression, markedly downregulated by scopolamine, was significantly upregulated by vibrotactile stimulation, with 4.81-fold and 4.04-fold increases in the V2.2 and V4 groups, respectively (Figure 10D). These results support the role of vibrotactile stimulation in modulating the AKT/GSK3β/β-catenin signaling axis.

Collectively, our results demonstrate that 40 Hz vibrotactile stimulation at 2.2 m/s^2^ yields the most robust neuroprotective effects across in vitro and in vivo models of neurotoxicity. This stimulation condition promoted neuronal differentiation, improved cognitive performance, restored cholinergic function, reduced oxidative stress and neuroinflammation, and enhanced synaptic plasticity. These findings support the therapeutic potential of vibrotactile stimulation as a non-invasive intervention for neurodegenerative disorders.

## 4. Discussion

The primary aim of this study was to investigate the acceleration-dependent effects of vibrotactile stimulation within the gamma-frequency range and to explore its potential as a non-invasive neuromodulation strategy. Our findings demonstrated that 2.2 m/s^2^ vibrotactile stimulation significantly improved cognitive function and attenuated scopolamine-induced neurotoxicity. The stimulation alleviated oxidative stress, cholinergic dysfunction, neuroinflammation, and apoptosis, while enhancing synaptic plasticity. These effects were associated with activation of the AKT/GSK3β/β-catenin signaling pathway and upregulation of neuronal and synaptic markers. Notably, the 2.2 m/s^2^ condition yielded significantly greater improvements than the 4.0 m/s^2^ condition in several key markers, including GAP43, PSD95, ChAT, and IL-1β (Appendix A), suggesting that moderate acceleration may be more effective for neuroprotection. Collectively, these results support the potential of vibrotactile stimulation as a non-invasive therapeutic approach for neurodegenerative diseases.

Gamma oscillations (25–100 Hz) generated by synchronized neuronal activity are essential for cognitive functions [49,50,51], especially memory processing in the hippocampus [52,53]. Abnormal gamma activity is associated with AD-related cognitive deficits [54,55,56], although the causal relationship remains debated. Recent studies show that sensory stimulation can entrain gamma rhythms [57], and techniques like GENUS synchronize brain networks using external stimuli [58]. When 40 Hz stimulation is repeatedly delivered, the sensory input entrains the corresponding sensory cortex, aligning brain oscillations to 40 Hz. This amplifies the brain’s natural gamma rhythms, which underlies the mechanism of GENUS. Photostimulation have been shown to effectively entrain gamma oscillations across widespread brain regions, reducing neuronal and synaptic loss and thereby ameliorating neuropathology. These approaches also attenuated inflammation and significantly reduced amyloid levels in the visual cortex, contributing to a decrease in plaque pathology [27,59,60]. Non-contact acoustic stimulation entrains 40 Hz gamma activity in the auditory cortex and hippocampus, leading to reductions in Aβ deposition, inhibition of GSK 3β, recovery of mitochondrial function, and improvements in cognitive performance [27,59]. Electromagnetic stimulation has been shown to induce gamma oscillations by modulating cortical excitability, enhancing cholinergic function, and restoring abnormal connectivity between the hippocampus and prefrontal cortex [23].

The SH-SY5Y cell line is widely utilized as a model for various neurodegenerative diseases, including AD [61]. Although SH-SY5Y cells possess dopaminergic characteristics in their undifferentiated state, RA-induced differentiation can promote cholinergic features [62,63]. However, we acknowledge that RA treatment alone does not fully recapitulate the phenotype of mature cholinergic neurons. Therefore, the current findings should be interpreted with caution, and future studies employing fully differentiated cholinergic neuron models may provide more physiologically relevant insights. Vibrotactile stimulation increased dendritic length and upregulated *MAP2*, *NeuroD1*, and *PSD95*, indicating enhanced neuronal maturation and synaptic development. These findings are consistent with previous studies showing similar effects on neuronal markers. Studies have reported that exposure to EMF increases dendritic length in SH-SY5Y cells [64]. Also, exposure of primary hippocampal neurons to EMF led to an upregulation of PSD95 and MAP2 expression [65]. Furthermore, the exposure of neural stem cells to EMF in the gamma frequency range was found to enhance *NeuroD1* gene expression [66]. These cellular changes are in line with our in vivo findings, where vibrotactile stimulation improved cognitive function and increased synaptic protein expression, suggesting that the upregulation of neuronal markers at the cellular level may contribute to synaptic restoration and functional recovery in the scopolamine-induced neurotoxicity mouse model. Only one of the three repeated experiments was included in the presentation of the data. We acknowledge that this approach may reduce statistical robustness, and recommend that future studies integrate multiple biological replicates to improve the reliability and generalizability of the findings.

Behaviorally, scopolamine-treated mice showed impairments in the MWM, NOR, and Y-maze tests, reflecting deficits in spatial, recognition, and working memory [48,67,68,69,70,71]. Previous reports showed improved NOR performance following photostimulation [36]. Additionally, 40 Hz non-contact acoustic stimulation significantly enhanced the object recognition index in the NOR test and markedly reduced escape latency in the MWM in 5XFAD mice [26]. Mice subjected to whole-body vibration exhibited a significant reduction in escape latency in the MWM test [72]. Consistent with these previous findings, in the MWM, scopolamine administration markedly increased escape latency, whereas vibrotactile stimulation significantly shortened latency, particularly under the 2.2 m/s^2^ condition. However, because MWM performance can be influenced by stress and motor ability, including potential changes induced by vibrotactile stimulation, baseline motor function was assessed prior to the experiment. In the NOR and Y-maze tests, scopolamine reduced novel object exploration and spontaneous alternation behavior, whereas vibrotactile stimulation improved measures.

Oxidative stress contributes to neuronal apoptosis and mitochondrial dysfunction in AD [73,74]. Elevated malondialdehyde (MDA) levels, a well-established biomarker of oxidative stress, are frequently observed in scopolamine models [75,76,77,78,79]. Exposure to EMF has been shown to reduce oxidative stress in rats [80], and transcranial magnetic stimulation (TMS) effectively decreased MDA levels in rats exposed to 3-nitropropionic acid-induced oxidative stress [81]. Consistent with these findings, our study observed a significant increase in MDA levels in the scopolamine-treated group. Moreover, donepezil effectively attenuated the scopolamine-induced elevation of MDA levels, consistent with previous reports [47]. These findings suggest that vibrotactile stimulation also may serve as an effective intervention for mitigating oxidative stress. Our previous work demonstrated that scopolamine-induced increases in MDA levels were more pronounced in the cortex than in the hippocampus [48]. In the present study, cortical tissues from multiple animals were pooled and reduced to two data points. This approach does not allow for direct assessment of inter-animal variability, which is particularly relevant given that oxidative stress markers can vary considerably between individuals. While we sought to represent the results using average values, future studies should perform individual analyses to provide more reliable and generalizable conclusions.

The disruption of cholinergic circuitry is closely associated with neurodegenerative diseases, including AD, and is a primary contributor to cognitive decline. ACh, a key neurotransmitter, is synthesized by ChAT and degraded by AChE [82]. Scopolamine administration has been shown to upregulate AChE while concurrently reducing ChAT levels [44,48,83,84,85]. AChE activity tends to decrease under EMF within the frequency range of gamma oscillation [86]. Additionally, whole-body vibration has been reported to enhance ChAT expression in the brains of C57Bl/6J mice [87]. Consistent with these previous findings, our study demonstrated that scopolamine administration led to an increase in AChE expression, while ChAT protein levels were significantly reduced.

Neurodegenerative diseases involve abnormal neuronal apoptosis, particularly in the hippocampus and cortex, where neuronal loss is closely linked to cognitive decline [88,89]. Apoptosis mediated by BAK and BAX disrupts mitochondrial function, contributing to neurodegeneration if unregulated [89,90]. Scopolamine has been shown to increase BAX expression in rats, which our study also confirmed through immunostaining and protein analysis [91,92]. Vibrotactile stimulation effectively attenuated this BAX upregulation, suggesting a neuroprotective role against scopolamine-induced neuronal death. This is consistent with previous findings that 40 Hz whole-body vibration reduces NeuN loss in aged rats [22] and decreases TUNEL-positive cells and cleaved caspase-1 expression in TBI models, thereby mitigating apoptosis [72].

A major hallmark of AD is the accumulation of senile plaques composed of Aβ, which induce oxidative stress and neuronal apoptosis [93,94]. Scopolamine has similarly been shown to increase Aβ burden, elevating APP mRNA and Aβ protein levels in rodent brains. Our findings align with this, demonstrating Aβ upregulation in scopolamine-treated mice. However, vibrotactile stimulation significantly reduced this increase. This is consistent with previous studies reporting that 40 Hz LED photostimulation and non-contact acoustic stimulation effectively decreased Aβ plaque load in 5XFAD mice [26].

Microglia are a major source of pro-inflammatory cytokines such as IL-1β, IL-4, and TNF-α in AD, contributing to neurotoxicity [95,96]. Scopolamine similarly increases these cytokines in rodent models [97], which we also observed in our study. This inflammatory response was significantly reduced by vibrotactile stimulation. Prior studies also support these effects, showing that 40 Hz vibration reduces neuroinflammation and that 40–80 Hz whole-body vibration at 1.01 m/s^2^ decreases hippocampal Iba1, IL-1β, and TNF-α levels in aged rats [22,98]. Since vibrotactile stimulation reduced inflammatory cytokines, these effects may indirectly contribute to synaptic restoration by alleviating the neuroinflammation-induced suppression of synaptic plasticity pathways. In particular, pro-inflammatory cytokines such as IL-1β and TNF-α are known to downregulate BDNF expression and interfere with TrkB signaling, thereby impairing synaptic maintenance and plasticity. Additionally, chronic inflammation promotes aberrant microglial activation, which can lead to excessive synaptic pruning and loss of dendritic spines. By reducing cytokine levels and microglial activation, vibrotactile stimulation may relieve these detrimental effects and facilitate the recovery of synaptic integrity and function [98,99,100].

BDNF is a key neurotrophic factor that regulates synaptic plasticity, neurogenesis, and neuroprotection, and its activation has been shown to reduce synaptic degeneration in AD models [101,102]. It is also known, along with NRG1, to facilitate gamma oscillations [103]. In earlier reports, vibrotactile stimulation upregulates synaptic plasticity-related proteins such as GAP43, PSD95, and SYN, which are involved in maintaining synaptic structure and transmission [104]. A whole-body vibration of 40 Hz at 1.01 m/s^2^ increased PSD95 expression in aged rats [22], and gamma-frequency EMF stimulation promoted neurodifferentiation in SH-SY5Y cells by elevating GAP43 levels [105]. These findings suggest that vibrotactile stimulation restores synaptic function compromised by scopolamine-induced neuroinflammation and neuronal loss. Notably, this is consistent with our in vitro observations, where stimulation increased neuronal and synaptic markers such as MAP2, NeuroD1, and PSD95 in Aβ-treated SH-SY5Y cells, suggesting a convergent mechanism of synaptic restoration at both the cellular and in vivo levels.

These molecular markers play a critical role in cognitive functions. In particular, the expression of ChAT is strongly associated with the regulation of learning and memory, and its reduction has been linked to cognitive decline [106,107]. Likewise, synaptic proteins such as PSD95, synaptophysin, and GAP43 are closely related to cortical atrophy and cognitive impairment in dementia [108,109]. Consistent with this, our results demonstrate that the group receiving vibrotactile stimulation at 2.2 m/s^2^ exhibited greater improvements in both cognitive performance and molecular marker expression compared to the 4.0 m/s^2^ group. Specifically, the recognition index in the NOR test increased by 1.4-fold in the V2.2 group and 1.2-fold in the V4 group relative to the scopolamine group. In parallel, synaptic and cholinergic markers showed more prominent upregulation in the V2.2 group: PSD95 increased by 1.58-fold in V2.2 vs. 0.57-fold in V4, GAP43 by 1.44-fold vs. 1.03-fold, and ChAT by 4.89-fold vs. 3.07-fold, respectively. These findings suggest that the cognitive enhancement observed with V2.2 stimulation may be proportionally linked to the restoration of molecular targets involved in synaptic plasticity and cholinergic function.

BDNF exerts neuroprotective effects by activating the AKT pathway, which suppresses Aβ-induced apoptosis via BAX downregulation and inactivates GSK-3β through Ser-9 phosphorylation [14,15,16]. GSK-3β activation is associated with reduced ACh levels and increased Aβ formation via BACE1 upregulation [17,18], while its inactivation stabilizes β-catenin, promoting Wnt/β-catenin signaling, which is critical for neuronal survival and synaptic plasticity [19,20]. Given its pivotal role in AD pathogenesis, the AKT/GSK-3β/β-catenin signaling pathway represents a crucial therapeutic target for AD treatment. Thus, the AKT/GSK-3β/β-catenin pathway is a key target in AD therapy. Supporting this, gamma-range EMF stimulation increased p-AKT and nuclear β-catenin levels in neural progenitor cells and enhanced cognition in mouse models [110,111,112]. In our study, scopolamine-induced reductions in BDNF and AKT/GSK-3β/β-catenin signaling were reversed by vibrotactile stimulation, with effects comparable to donepezil. These findings suggest that vibrotactile stimulation promotes BDNF-mediated activation of AKT signaling, contributing to neuroprotection and cognitive improvement.

The neuroprotective effects of vibrotactile stimulation are strongly influenced by specific stimulation parameters, such as acceleration, frequency, and duration. Importantly, these parameters do not act independently but rather in a synergistic and interdependent manner. Their combined effects determine the activation threshold and dynamics of mechanosensory receptors, including Piezo1 and TRP channels, which, in turn, modulate downstream signaling pathways such as AKT/GSK3β and ERK/CREB [31,113,114,115,116,117,118]. This complex interplay can influence a wide range of biological processes, including neuroinflammation, synaptic plasticity, and cellular resilience. As such, both the molecular mechanisms and functional outcomes may vary depending on the specific combination of stimulation parameters. Consequently, even subtle changes in stimulation conditions may lead to markedly different neuroprotective and cognitive effects, potentially explaining the variability observed across studies and disease models [31,119].

In conclusion, this study demonstrates that 40 Hz vibrotactile stimulation, particularly at 2.2 m/s^2^, exerts significant neuroprotective and cognitive benefits through modulation of oxidative stress, inflammation, apoptosis, and synaptic function via the AKT/GSK3β/β-catenin pathway. These results suggest that optimal acceleration is critical for therapeutic efficacy and highlight the potential of vibrotactile stimulation as a non-invasive neuromodulation strategy. However, limitations include the use of a scopolamine-induced model that does not fully replicate Alzheimer’s disease pathology, reliance on SH-SY5Y cells lacking mature cholinergic phenotypes, and the absence of long-term or sex-specific analyses. Future studies should address these issues using chronic models, both sexes, and extended stimulation protocols to further validate the translational potential of this approach.

## 5. Conclusions

This study highlights the therapeutic potential of vibrotactile stimulation as a non-invasive neuromodulation strategy for mitigating cognitive deficits in neurodegenerative conditions. By comparing different stimulation intensities, we identified 2.2 m/s^2^ as the optimal acceleration for promoting cognitive recovery in a scopolamine-induced neurotoxicity mouse model. Stimulation at this parameter significantly alleviated oxidative stress, neuroinflammation, cholinergic dysfunction, and apoptosis, while enhancing synaptic plasticity and cognitive performance. These effects were supported by molecular evidence showing the activation of the AKT/GSK-3β/β-catenin signaling pathway and the upregulation of neuronal and synaptic markers. Importantly, our findings underscore the relevance of stimulation parameter optimization in determining therapeutic efficacy. From a translational perspective, vibrotactile stimulation offers a promising, non-pharmacological approach that aligns with ongoing efforts in sensory-based interventions such as GENUS. However, further research is required to validate the long-term safety, stimulation thresholds, and feasibility of clinical application. Future studies should also explore its effectiveness across different models of neurodegeneration and examine the interplay of other parameters such as frequency, duration, and stimulation site to refine its therapeutic impact.

## Figures and Tables

**Figure 1 biomedicines-13-02031-f001:**
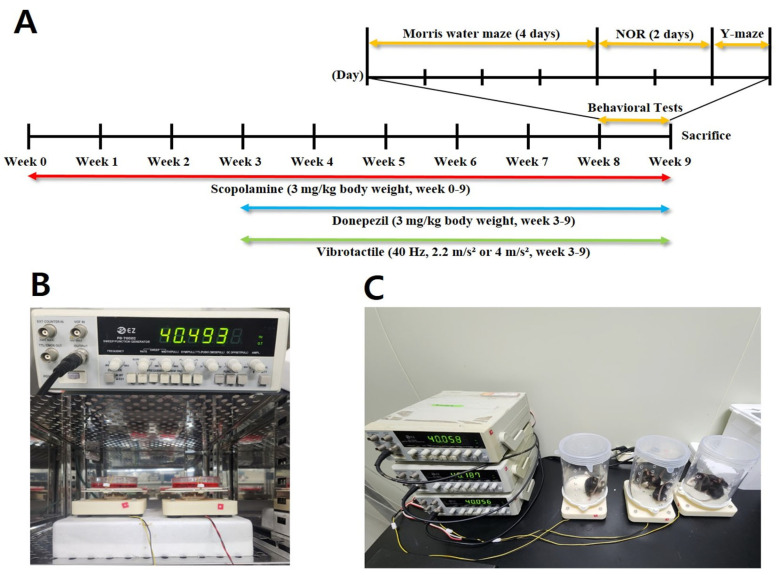
Vibrotactile stimulation setup. (**A**) Experimental timeline illustrating treatment and testing schedule. (**B**) Setup for applying vibrations to cells using a 40 Hz sinusoidal waveform from a function generator. The 100 mm dish was secured with rubber bands for stable stimulation. (**C**) Setup for vibrotactile stimulation in mice. A cylindrical plastic barrier was placed around the vibration plate to prevent mice from leaving. A single vibrator accommodated up to four mice simultaneously.

**Figure 2 biomedicines-13-02031-f002:**
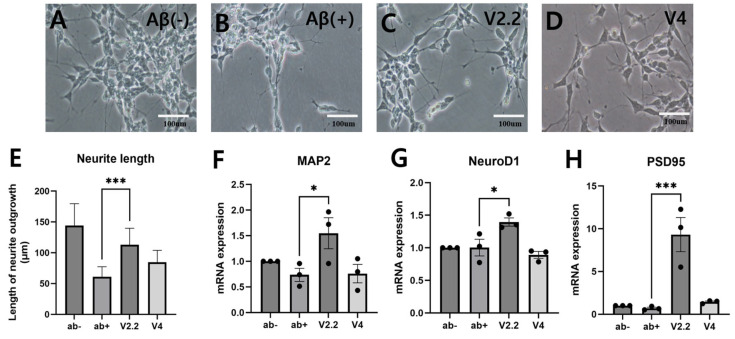
Effect of vibrotactile stimulation on SH-SY5Y Cells. (**A**–**D**) Representative images showing the morphology of SH-SY5Y cells (100× magnification). Scale bar = 100 μm. (**E**) Quantification of neurite length in differentiated SH-SY5Y cells was performed using ImageJ. (**F**–**H**) mRNA expression levels of *MAP2* (**F**), *NeuroD1* (**G**), and *PSD95* (**H**) were quantified using RT-qPCR under the respective conditions. Statistical significance is indicated as follows: * *p* < 0.05, *** *p* < 0.005. All statistical comparisons were performed using one-way ANOVA followed by Tukey’s multiple comparisons test. As control group (Aβ−) values were normalized to 1, all biological replicates showed identical values, and therefore no error bars are visible for this group.

**Figure 3 biomedicines-13-02031-f003:**
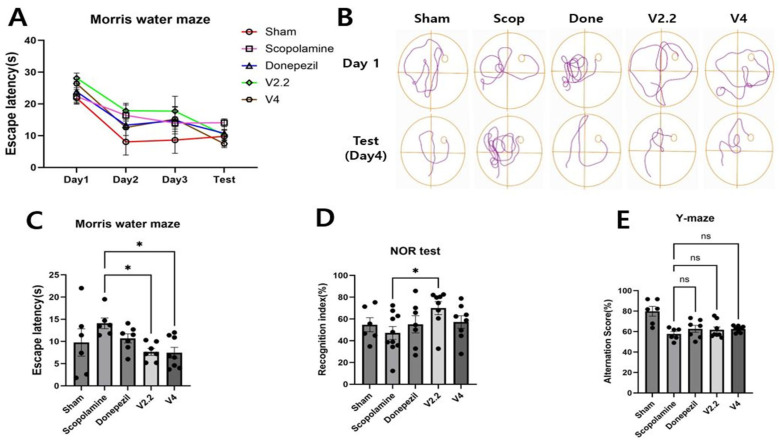
Effects of vibrotactile stimulation on spatial learning and memory in a scopolamine-induced neurotoxicity mouse model. (**A**–**C**) Morris water maze results showing escape latency trends over 4 days (**A**), representative swim paths of mice on Day 1 and test day (**B**), and escape latency on test day (**C**). (**D**) Novel object recognition (NOR) test results represented as recognition index (%). (**E**) Y-maze results showing alternation score (%). Data are presented as mean ± SEM. Symbols * and ** indicate significant differences at *p* < 0.05 and *p* < 0.01, respectively; ns indicates not significant. Each group included 10 animals (n = 10). Behavioral experiments represent biological replicates. All statistical comparisons were performed using one-way ANOVA followed by Tukey’s multiple comparisons test.

**Figure 4 biomedicines-13-02031-f004:**
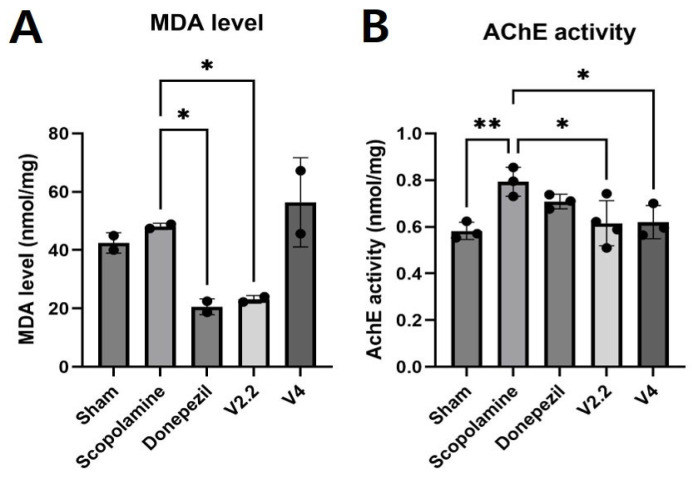
Effects of vibrotactile stimulation on reducing oxidative stress and AChE activity in a scopolamine-induced neurotoxicity mouse model. (**A**) Malondialdehyde (MDA) levels in the cortex, an indicator of lipid peroxidation and oxidative stress. (**B**) Acetylcholinesterase (AChE) activity in the cortex, which is associated with cholinergic neurotransmission. Data are presented as mean ± SEM. Statistical significance is indicated as follows: * *p* < 0.05, ** *p* < 0.01. Each group included four animals (n = 4). These data represent biological replicates.

**Figure 5 biomedicines-13-02031-f005:**
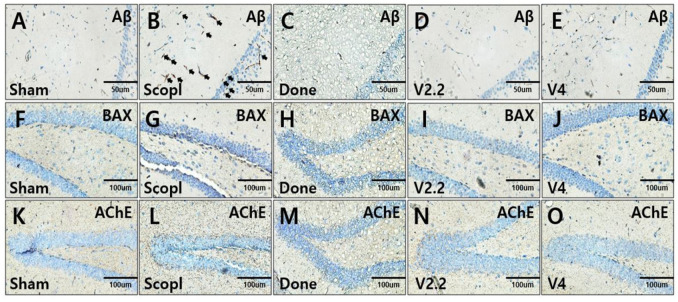
Immunohistochemical analysis of Aβ, BAX, and AChE expression in hippocampus of a scopolamine-induced neurotoxicity mouse model. (**A**–**E**) Aβ (Amyloid-β) expression in the hippocampus. (**B**) Arrows indicate Aβ-positive cells. (**F**–**J**) BAX, a pro-apoptotic protein involved in neuronal cell death, expression in the hippocampus. (**K**–**O**) AChE expression in the hippocampus. Scale bars: 50 μm (**A**–**E**), 100 μm (**F**–**J**).

**Figure 6 biomedicines-13-02031-f006:**
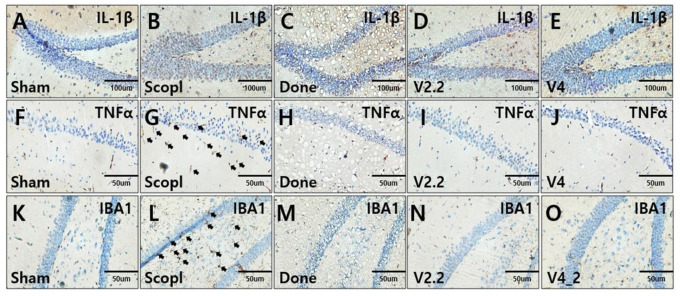
Immunohistochemical analysis of inflammatory markers IL-1β, TNF-α, and IBA1 expression in hippocampus of a scopolamine-induced neurotoxicity mouse model. (**A**–**E**) IL-1β expression in the hippocampus. (**F**–**J**) TNF-α expression in the hippocampus. (**K**–**O**) IBA1 expression in the hippocampus. IL-1β and TNF-α are pro-inflammatory cytokines, while IBA1 is a marker for microglial activation, indicating neuroinflammation. Arrows indicate TNFα-positive cells (**G**) and IBA1-positive microglia (**L**). Scale bars: 100 μm (**A**–**E**), 50 μm (**F**–**O**). An overview of hippocampal immunostaining can be found in Appendix A.

**Figure 7 biomedicines-13-02031-f007:**
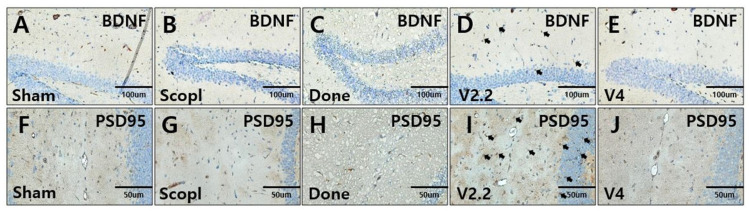
Immunohistochemical analysis of neuronal markers BDNF, PSD-95 expression in hippocampus of a scopolamine-induced neurotoxicity mouse model. (**A**–**E**) BDNF expression in the hippocampus. (**F**–**J**) PSD-95 expression in the hippocampus. Arrows indicate BDNF-positive cells (**D**) and PSD95-positive neurons (**I**). Scale bars: 100 μm (**A**–**E**,**K**–**O**), 50 μm (**F**–**J**). An overview of hippocampal immunostaining can be found in Appendix A.

**Figure 8 biomedicines-13-02031-f008:**
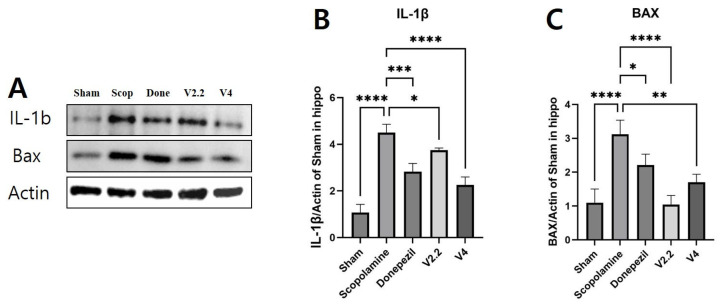
Western blot analysis of IL-1β and BAX expression in hippocampus of a scopolamine-induced neurotoxicity mouse model. (**A**) Representative Western blot images showing expression levels of IL-1β, BAX, and Actin. (**B**) Quantification of IL-1β expression levels normalized to Actin. (**C**) Quantification of BAX expression levels normalized to Actin. (4 m/s^2^) group (V4). Data are presented as mean ± SEM. Statistical significance is indicated as * *p* < 0.05, ** *p* < 0.01, *** *p* < 0.001, **** *p* < 0.0001. All statistical comparisons were performed using one-way ANOVA followed by Tukey’s multiple comparisons test. These data represent technical replicates. β-actin panel represents common loading control for all target proteins because loading conditions were identical across experiments. Full-length, uncropped images of all Western blots shown in this figure are provided in Appendix A.

**Figure 9 biomedicines-13-02031-f009:**
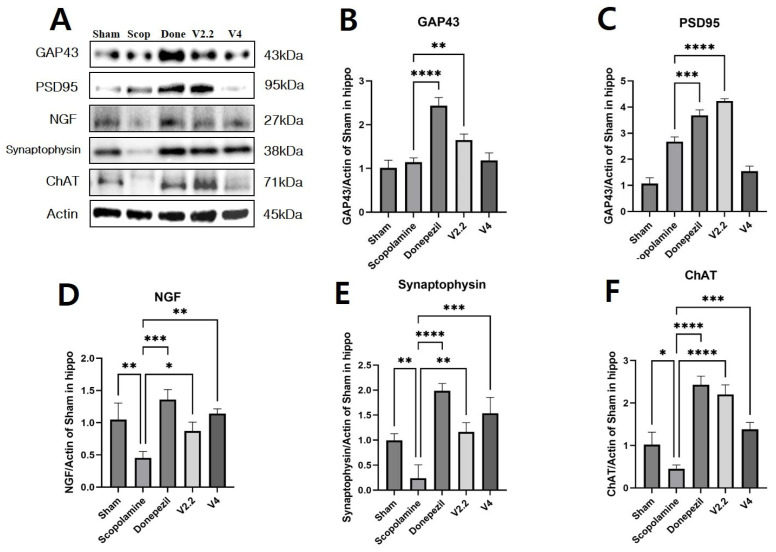
Western blot analysis of neural markers in the hippocampus of a scopolamine-induced neurotoxicity mouse model. (**A**) Representative Western blot images showing expression levels of GAP43, PSD95, NGF, Synaptophysin, ChAT, and Actin in the hippocampus. (**B**–**F**) Quantification of GAP43, PSD95, NGF, Synaptophysin, and ChAT expression levels normalized to Actin. Data are presented as mean ± SEM. Statistical significance is indicated as * *p* < 0.05, ** *p* < 0.01, *** *p* < 0.001, **** *p* < 0.0001. All statistical comparisons were performed using one-way ANOVA followed by Tukey’s multiple comparisons test. These data represent technical replicates. β-actin panel represents common loading control for all target proteins because loading conditions were identical across experiments. Full-length, uncropped images of all Western blots shown in this figure are provided in Appendix A.

**Figure 10 biomedicines-13-02031-f010:**
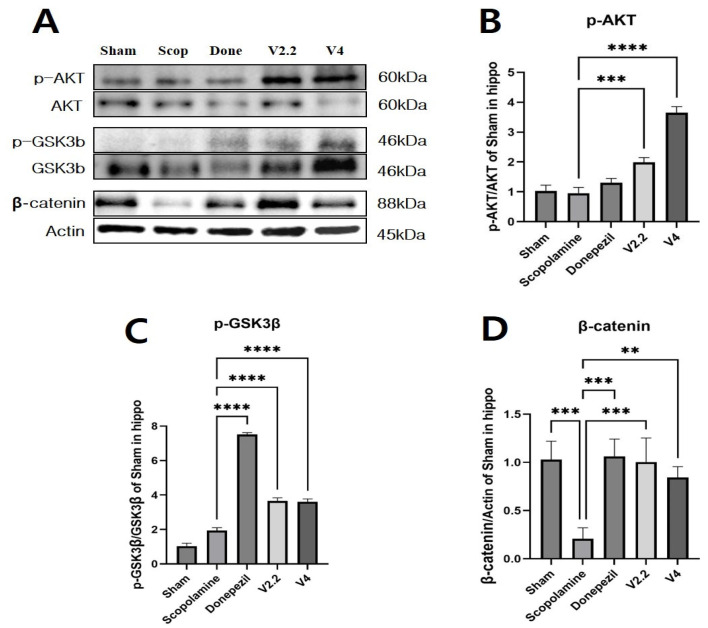
Western blot analysis of AKT/GSK3β/β-catenin in hippocampus of a scopolamine-induced neurotoxicity mouse model. (**A**) Representative Western blot images showing expression levels of phosphorylated AKT (p-AKT), total AKT, phosphorylated GSK3β (p-GSK3β), total GSK3β, β-catenin, and Actin (loading control) in the hippocampus. (**B**–**D**) Data are presented as mean ± SEM. Statistical significance is indicated as ** *p* < 0.01, *** *p* < 0.005, **** *p* < 0.001. All statistical comparisons were performed using one-way ANOVA followed by Tukey’s multiple comparisons test. These data represent technical replicates.

## Data Availability

Data are contained within the article.

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
