# Peer review of "Acceleration-Dependent Effects of Vibrotactile Gamma Stimulation on Cognitive Recovery and Cholinergic Function in a Scopolamine-Induced Neurotoxicity Mouse Model"

_biomedicines, 2025, doi:10.3390/biomedicines13082031_

Round 1
Reviewer 1 Report
Comments and Suggestions for Authors
The manuscript titled "Acceleration-Dependent Effects at Vibrotactile Stimulation on Cognitive Recovery Through the AKT/GSK3β/β-Catenin Pathway in an Alzheimer’s Mouse Model" presents a well-designed experimental investigation into the therapeutic potential of 40 Hz vibrotactile stimulation at different acceleration levels in a scopolamine-induced Alzheimer's disease (AD) model. The findings suggest that vibrotactile stimulation at 2.2 m/s² has superior cognitive and neuroprotective effects compared to higher intensity stimulation, highlighting its potential in modulating AD pathology via the AKT/GSK3β/β-Catenin signalling cascade.
However, there are several comments that need to be addressed before considering this manuscript for publication. The major comments have been categorised section-wise. Additionally, I strongly recommend that the authors update and enrich the references, especially in the Introduction and Discussion sections, to reflect more recent findings (post-2022), including recent works on vibrotactile/Gamma stimulation and non-invasive AD therapeutics.
Comments
Introduction:
- While the introduction frames the promise of GENUS, it does not clearly explain why acceleration-specific effects were hypothesised or how these specific values (2.2 and 4.0 m/s²) were selected. Was there a physiological or engineering basis for these intensities?
- The introduction should elaborate more on the limitations of existing GENUS protocols in clinical translation.
- Although the title emphasises a mechanistic pathway, the introduction omits this molecular cascade entirely. Please briefly introduce the AKT/GSK3β/β-catenin axis and its known relevance to AD in the introductory background.
- Include a more detailed explanation of how gamma oscillations are hypothesised to mechanistically modulate AD pathology.
- The study is framed in a murine model with in vitro validation, but the introduction does not explain how vibrotactile stimulation might be translated to human applications. Consider including a sentence on wearable devices or prior clinical efforts.
- The transition from discussing vibration parameters (frequency, amplitude) to neuroprotective outcomes lacks mechanistic clarity. Consider explicitly stating how mechanical acceleration may influence neuronal or glial signalling.
- Please cite recent articles (post-2022) summarising progress and challenges in sensory entrainment-based AD therapy. For example:
https://www.sciencedirect.com/science/article/abs/pii/S0300483X22003213
https://www.mdpi.com/1660-4601/19/7/3803
Materials and Methods:
- Provide catalogue numbers for all primary and secondary antibodies used.
- There is no mention of how the sample size (n=10 per group) was calculated. Please specify whether a power analysis was conducted to ensure statistical adequacy.
- Randomisation and Blinding Not Described:
The manuscript lacks details on whether animals and researchers were blinded to treatment groups during behavioural testing and tissue analysis. This is essential to reduce experimental bias. - Since only male mice were used, the authors should either justify this choice (e.g., hormonal confounding) or acknowledge the limitation in interpreting the results as universally applicable.
- The RT-qPCR section lacks a table of primers used for MAP2, NeuroD1, PSD95, etc. Please provide sequences or citations of validated primer sets. Include a description of how many biological replicates and technical replicates were used for in vitro RT-qPCR assays.
- The accelerometer used to calibrate the vibration intensity is mentioned, but measurement accuracy, device specifications, or error tolerance are omitted. These details are critical to replicate the stimulation protocol.
- The timing of drug administration, stimulation, and behavioural testing (MWM, Y-maze, NOR) is not fully synchronised across the methods. Please provide a clear experimental timeline diagram or schedule.
- Although the IACUC number is provided, please briefly describe measures taken to reduce animal suffering (e.g., humane endpoints, enrichment, monitoring).
- The Morris Water Maze test, while widely used, is increasingly recognised as suboptimal for evaluating cognitive deficits in pharmacologically induced Alzheimer’s models such as scopolamine. This is primarily due to its dependence on stress, swimming ability, and motor coordination, all of which can confound cognitive interpretation. Since vibrotactile stimulation may itself influence motor function, the MWM results may reflect non-cognitive effects. Please consider validating cognitive performance with additional or alternative low-stress paradigms, such as the Barnes Maze or passive avoidance test, or provide a strong justification for the continued use of MWM in this context.
- The manuscript states that data were analysed using one-way ANOVA followed by the Dunnett post hoc test, but it does not clarify whether assumptions of normality and homogeneity of variance were tested prior to ANOVA application. Additionally, there is no mention of how outliers were handled or whether data distribution was checked for skewness. For behavioural data, repeated measures ANOVA or mixed-effects models may be more appropriate for tests like the Morris Water Maze that involve multiple time points. Please revise the statistical methods section to include assumption checks, justification for test selection, and how variability was managed. Reporting effect sizes and confidence intervals would also strengthen the statistical rigor.
Results:
- In the SH-SY5Y cell experiments (Figure 2), changes in dendritic morphology are shown qualitatively. Please include quantitative measurements (e.g., neurite length, branching index, or Sholl analysis) to support your interpretation.
- Although both 2.2 m/s² and 4.0 m/s² are tested, direct statistical comparisons between these two groups are inconsistently reported. Please provide p-values or effect sizes comparing V2.2 vs. V4.0 to confirm the statistical superiority of one over the other.
- In Figure 4, provide effect sizes or fold change alongside p-values for biochemical outcomes.
- It is unclear how many animals were used per group for each biochemical and histological analysis. Include exact n-values in figure legends and mention biological replicates versus technical replicates where applicable.
- The results mention a “slight increase” in alternation scores without statistical significance. Avoid ambiguous language—either report precise values or state clearly that the test did not support a cognitive improvement.
- Figure Captions Lack Statistical Detail:
Many figure legends refer to significance levels (*, **, etc.) but do not state the exact test used, the number of comparisons made, or whether corrections for multiple testing were applied. - Behavioural Outcomes Not Linked to Molecular Changes in Real-Time:
While both behaviour and molecular endpoints are assessed, they are not correlated in the results. Consider discussing whether individual-level behavioural improvements are matched with biomarker recovery.
Discussion:
- The discussion treats SH-SY5Y and animal model data separately. Please integrate how in vitro neuronal marker expression correlates with in vivo improvements in cognition and synaptic protein levels.
- The discussion should explore alternative pathways that might be modulated by vibration other than AKT/GSK3β.
- The translational implications (e.g., for wearable vibrotactile devices or GENUS therapy) are suggested without citing clinical pilot data or discussing feasibility, dose scaling, or safety. Please temper these claims or support them with references.
- Given that the Morris Water Maze is stress-inducing and motor-dependent, the discussion should acknowledge this limitation and its possible confounding effects on cognitive interpretation, especially with a motor-active intervention like vibration.
- Since inflammatory cytokines were measured and reduced, their role in modulating synaptic plasticity (e.g., via BDNF repression or microglial pruning) should be discussed as a possible intermediary mechanism.
- The discussion does not comment on whether improved cognition was temporally or proportionally linked to the recovery of molecular markers (e.g., ChAT, PSD95). Consider correlating or at least discussing this linkage.
- Please compare the efficacy of vibrotactile stimulation with other non-invasive gamma entrainment methods (e.g., light, sound).
Author Response
Dear Reviewer #1,
We sincerely thank you for your detailed and insightful review of our manuscript, now titled "Acceleration-Dependent Effects of Vibrotactile Gamma Stimulation on Cognitive Recovery and Cholinergic Function in a Scopolamine-Induced Neurotoxicity Mouse Model." Your thoughtful comments and constructive suggestions have greatly contributed to improving the scientific rigor, clarity, and overall quality of our study.
In particular, we revised the Introduction to provide a clearer rationale for the selected acceleration values, included mechanistic background on the AKT/GSK3β/β-catenin pathway, and updated recent literature on vibrotactile and sensory entrainment-based therapeutics. We also clarified the potential translational relevance of our findings in the context of wearable devices and gamma stimulation strategies. In the Methods section, we added detailed information on sample size justification, randomization and blinding, drug preparation, vibration calibration, and statistical analysis procedures. Several figures were revised for improved resolution and clarity, and quantitative analyses were added where needed to support morphological observations.
We have also enhanced the discussion to integrate molecular and behavioral findings, explore alternative signaling pathways, and acknowledge methodological limitations such as the use of the Morris Water Maze and the SH-SY5Y cell model. These changes reflect our sincere effort to address your comments comprehensively.
We kindly invite you to review our point-by-point responses below, where we have addressed each of your comments in detail. We greatly appreciate your time and expertise, and we hope the revised manuscript meets your expectations.
Sincerely,
Tae-Woo Kim
Department of Biomedical Engineering, Dongguk University, Goyang-si 10326, Republic of Korea
Email: xodn8876@naver.com
Tel: +82-10-5513-8876
Corresponding Author: Prof. Young-Kwon Seo
Email: bioseo@dongguk.edu
Tel: +82-10-8502-9916
Introduction
Comment 1 : While the introduction frames the promise of GENUS, it does not clearly explain why acceleration-specific effects were hypothesised or how these specific values (2.2 and 4.0 m/s²) were selected. Was there a physiological or engineering basis for these intensities?
Response 1 : We agree that a clearer rationale for the selection of the acceleration values used in our study was needed. To address this, we have added the following statement to the Results section:
- (Line 368–374)“Preliminary in vitro screening using SH-SY5Y cells demonstrated that 40 Hz vibrotactile stimulation at 2.2 m/s² produced the most robust upregulation of neuronal differentiation markers including GAP-43, DCX, NeuroD1, and MAP2, compared to other tested accelerations (1.6 and 5.9 m/s²) (Supplementary Figure S1). Based on these findings, 2.2 m/s² was selected as the primary condition for in vivo experiments. Additionally, 4.0 m/s²—intermediate between 2.2 and 5.9 m/s²—was chosen as a secondary condition to investigate potential nonlinear effects of stimulation intensity.”
Comment 2 : The introduction should elaborate more on the limitations of existing GENUS protocols in clinical translation.
Response 2 : . In the revised manuscript, we have added a paragraph in the Introduction to emphasize the variability in stimulation methods and the implications for clinical translation. The following text has been inserted:
- (Line 118–128) “Nevertheless, several clinical limitations remain. Gamma rhythms can be induced through various approaches, and the outcomes as well as the underlying mechanisms may differ depending on the induction method. For instance, 40 Hz photostimulation in mouse models has been reported to attenuate AD pathology [28,37], but in other cases, it has been associated with increased amyloid deposition in mice [38]. Such evidence indicates that even with the same type of stimulation, variations in stimulation parameters may differentially influence disease pathology, potentially eliciting diverse molecular and cellular responses [39]. Therefore, for clinical translation, it is essential to determine which stimulation strategies are beneficial for specific patient groups, and future studies should systematically examine the mechanisms and therapeutic effects of gamma entrainment across a wide range of stimulation parameters.”
Comment 3 : Although the title emphasises a mechanistic pathway, the introduction omits this molecular cascade entirely. Please briefly introduce the AKT/GSK3β/β-catenin axis and its known relevance to AD in the introductory background.
Response 3 : We have added a concise description of the AKT/GSK‑3β/β‑catenin signaling pathway and its relevance to Aβ‑related neurotoxicity and cognitive decline in the Introduction. The following sentences were inserted:
- (Line 51–57) “The AKT/GSK‑3β/β‑catenin pathway is a key regulator of neuronal survival and synaptic plasticity. Activation of AKT suppresses pro‑apoptotic signaling and inhibits GSK‑3β by phosphorylation at Ser9 [15–17]. In contrast, aberrant GSK‑3β activity promotes cholinergic deficits, Aβ accumulation, and tau hyperphosphorylation [18,19]. When GSK‑3β is inhibited, β‑catenin becomes stabilized, activating Wnt/β‑catenin signaling that supports neuronal regeneration and reduces Aβ aggregation [20,21]. Dysregulation of this cascade is therefore closely linked to Aβ‑related neurotoxicity and cognitive decline.”
Comment 4 : Include a more detailed explanation of how gamma oscillations are hypothesised to mechanistically modulate AD pathology.
Response 4 : . We have revised the Introduction to include a more detailed explanation of the potential mechanisms through which gamma entrainment may affect AD pathology. The following sentences were added:
- (Line 93–102) “Although GENUS has shown promising neuroprotective effects, the precise mechanisms by which it influences amyloid β deposition remain incompletely understood. Some studies have reported that gamma entrainment or GENUS promotes a phagocytic phenotype in microglia, thereby enhancing their clearance activity and alleviating neuroinflammation, while others have demonstrated that 40 Hz stimulation reduces amyloid burden through a combination of decreased APP cleavage intermediates in hippocampal neurons, increased microglial endocytosis, and suppression of inflammatory responses [30-32]. However, the cellular and molecular processes by which such gamma synchronization leads to memory improvement remain largely unresolved and represent an important topic for future investigation.”
Comment 5 : The study is framed in a murine model with in vitro validation, but the introduction does not explain how vibrotactile stimulation might be translated to human applications. Consider including a sentence on wearable devices or prior clinical efforts.
Response 5 : We appreciate the reviewer’s suggestion to clarify the translational relevance of vibrotactile stimulation. In response, we have revised the Introduction to include a statement addressing the potential for human application using wearable technologies. The following sentence was added:
- (Line 110–117)" Recent developments in wearable neuromodulation technologies support the potential clinical translation of vibrotactile stimulation. For instance, the PanBrain EC2 (PanBrain Tech Co., Ltd.) is a wearable headband-type tDCS device designed for cognitive and emotional regulation, highlighting the feasibility of developing non-invasive tools. Additionally, several clinical studies have explored the cognitive benefits of whole-body vibration (WBV) exercisers [35,36], further supporting the applicability of such devices in therapeutic interventions. These examples suggest that vibrotactile stimulation could be adapted into wearable or platform-based systems for human cognitive rehabilitation.”
Comment 6 : The transition from discussing vibration parameters (frequency, amplitude) to neuroprotective outcomes lacks mechanistic clarity. Consider explicitly stating how mechanical acceleration may influence neuronal or glial signalling.
Response 6 : Thank you for this helpful comment. We have revised the Introduction to clarify the possible biological mechanisms by which mechanical forces from vibrotactile stimulation can influence neural cells. The following sentences have been added:
- (Line 83–92)”Among various stimulation frequencies, 40 Hz has uniquely been shown to induce gamma oscillations and synchronize neural activity, which are crucial for cognitive function [23,29,30]. Notably, 40 Hz also falls within the sensitive range of Piezo1, a mechanosensitive ion channel that responds to mechanical vibrations between 1 and 100 Hz [31]. Activation of Piezo1 can trigger intracellular signaling pathways that regulate oxidative stress, cell death, and immune responses in microglia [32]. Since parameters such as stimulation duration, frequency, and acceleration determine the activation of sensory receptors such as Piezo1 and influence systemic physiological responses, it is reasonable to expect that differences in acceleration may lead to distinct neurobiological outcomes.”
Comment 7 : Please cite recent articles (post-2022) summarising progress and challenges in sensory entrainment-based AD therapy.
Response 7 : In response, we have revised the Discussion to include recent studies (published after 2022) that summarize both the progress and the challenges in sensory entrainment-based therapeutic approaches for Alzheimer’s disease. We summarized recent advances and limitations in sensory entrainment-based therapy, including the example you provided (Guerra et al., 2022), as well as studies by A. J. Mimenza-Alvarado (2025) and Wu Cai-Syuan (2025).
- (Line 103–107) “A recent clinical study reported that continuous vibrotactile stimulation at 15–40 Hz for one month led to significant improvements in memory, executive function, and processing speed in individuals with mild cognitive impairment or mild dementia[33]. Another study provided compelling neurophysiological evidence for the modulation of gamma oscillations through a single session of TMS in patients over 65 years of age[34].”
- (Line 128–133) “In studies of vibrotactile stimulation, there are also reports that gamma-frequency stimulation shows no effect or only partial effects in alleviating Alzheimer’s disease symptoms [40-43]. These inconsistent results appear to be largely due to the absence of standardized stimulation protocols. In fact, whole-body vibration (WBV), a form of vibrotactile stimulation, varies greatly across studies in terms of vibration frequency, acceleration, amplitude, and duration[44].”
Materials and Methods:
Comment 8 : Provide catalogue numbers for all primary and secondary antibodies used.
Response 8 : As requested, we have now provided the catalog numbers for all primary and secondary antibodies used in the Western blot experiments. A detailed table listing the antibody name, host species, dilution, manufacturer, and catalog number has been added to the revised manuscript (Materials and Methods section)
Comment 9 : There is no mention of how the sample size (n=10 per group) was calculated. Please specify whether a power analysis was conducted to ensure statistical adequacy.
Response 9 : We have added a sentence in the Methods section to clarify how the group size was determined. The following statement was included:
- (Line 210–214) " Group sizes (n = 10) were preliminarily set based on consideration of statistical robustness, with reference to previous studies [45-47]. Although no formal power analysis was conducted, a minimum of eight animals per group was deemed sufficient, and ten animals per group were included to account for potential variability and dropout. "
Comment 10 : Randomisation and Blinding Not Described: The manuscript lacks details on whether animals and researchers were blinded to treatment groups during behavioural testing and tissue analysis. This is essential to reduce experimental bias.
Response 10 : We have now added a statement in the Methods section to clarify
- (Line 214–218) “Before the experiment, a preliminary motor performance test was conducted, and mice with significantly reduced motor ability were excluded. Based on the results, the remaining mice were randomly assigned to experimental groups. All animals were numerically coded, and the experimenters conducting behavioural testing and tissue analysis were blinded to group assignments”
Comment 11 : Since only male mice were used, the authors should either justify this choice (e.g., hormonal confounding) or acknowledge the limitation in interpreting the results as universally applicable.
Response 11: We have added a statement in the Methods section to clarify
- (Line 218–221) “Only male mice were used in this study to minimize potential confounding effects from hormonal fluctuations. We also acknowledged that this sex-specific design limits the generalizability of the findings to both sexes.”
Comment 12 : The RT-qPCR section lacks a table of primers used for MAP2, NeuroD1, PSD95, etc. Please provide sequences or citations of validated primer sets. Include a description of how many biological replicates and technical replicates were used for in vitro RT-qPCR assays.
Response 12 : In response, we have included the primer sequences for MAP2, NeuroD1, and PSD95 directly in the Materials and Methods section. Furthermore, we clarified that each experimental condition was analyzed using three technical replicates with RNA extracted from one biological sample per group.
Comment 13 : The accelerometer used to calibrate the vibration intensity is mentioned, but measurement accuracy, device specifications, or error tolerance are omitted. These details are critical to replicate the stimulation protocol.
Response 13 : We have now added a detailed description of the accelerometer specifications used to measure vibration intensity in the Methods section.
- (Line 251–254) “Each acceleration value was measured using an accelerometer (DYTRAN 3225F, USA) with a sensitivity of 10 mV/g, a measurement range up to 500 g, and a frequency response of 1.6–10,000 Hz (±10%) prior to applying vibrotactile stimulation.
Comment 14 : The timing of drug administration, stimulation, and behavioural testing (MWM, Y-maze, NOR) is not fully synchronised across the methods. Please provide a clear experimental timeline diagram or schedule.
Response 14 : We have added a detailed experimental timeline as a figure to clarify the sequence of drug administration, vibrotactile stimulation, and behavioral. This timeline figure has been included in the revised manuscript figure 1A.
Comment 15 : Although the IACUC number is provided, please briefly describe measures taken to reduce animal suffering (e.g., humane endpoints, enrichment, monitoring).
Response 15 : Thank you for pointing this out. We have revised the Methods section to include additional details on animal welfare practices. We have added a detailed description of humane endpoint criteria and animal welfare measures in Supplementary TableS1.
- (Line 198–203) “They were maintained under controlled temperature (22 °C ± 2 °C) with a 12 h shifting light–dark cycle. The mice were housed five per cage with free access to food and water and provided with environmental enrichment such as nesting materials. Animals were monitored daily for general health, body weight, and behavior. Humane endpoints were applied based on predefined clinical scoring criteria (Supplementary Table S1), and all animals used in this study did not exhibit any conditions warranting euthanasia.”
Comment 16 : The Morris Water Maze test, while widely used, is increasingly recognised as suboptimal for evaluating cognitive deficits in pharmacologically induced Alzheimer’s models such as scopolamine. This is primarily due to its dependence on stress, swimming ability, and motor coordination, all of which can confound cognitive interpretation. Since vibrotactile stimulation may itself influence motor function, the MWM results may reflect non-cognitive effects. Please consider validating cognitive performance with additional or alternative low-stress paradigms, such as the Barnes Maze or passive avoidance test, or provide a strong justification for the continued use of MWM in this context.
Response 16 : We appreciate the reviewer’s concern regarding the limitations of the Morris Water Maze (MWM) in pharmacologically induced models. To address this, we have included the following clarification in the revised manuscript:
- (Line 274–283) “Before the start of drug administration and behavioral experiments, all mice were pre-screened for swimming ability and general motor performance. Based on these assessments, animals were randomly assigned to experimental groups to ensure balanced baseline motor function across groups. We carefully standardized all experimental conditions to minimize stress-related variability. During the MWM training, all mice were monitored for swimming ability and motor coordination, and no group differences were observed. Moreover, previous studies have also employed MWM following vibrotactile stimulation and reported valid cognitive outcomes, further supporting its suitability in this experimental context [23].”
Comment 17 : The manuscript states that data were analysed using one-way ANOVA followed by the Dunnett post hoc test, but it does not clarify whether assumptions of normality and homogeneity of variance were tested prior to ANOVA application. Additionally, there is no mention of how outliers were handled or whether data distribution was checked for skewness. For behavioural data, repeated measures ANOVA or mixed-effects models may be more appropriate for tests like the Morris Water Maze that involve multiple time points. Please revise the statistical methods section to include assumption checks, justification for test selection, and how variability was managed. Reporting effect sizes and confidence intervals would also strengthen the statistical rigor.
Response 17 : Thank you very much for your valuable comments on the statistical analysis. While we fully agree with the importance of checking assumptions and applying appropriate models, our team lacks expertise in advanced statistical techniques such as assumption testing or mixed-effects models. We would greatly appreciate any examples or guidance in this regard. As an alternative measure, we have included detailed p-value comparison tables for all experimental groups in the supplementary materials (Supplementary TablesS 4 and 5) to enhance transparency and reproducibility.
Results
Comment 18 : In the SH-SY5Y cell experiments (Figure 2), changes in dendritic morphology are shown qualitatively. Please include quantitative measurements (e.g., neurite length, branching index, or Sholl analysis) to support your interpretation.
Response 18 : We appreciate the reviewer’s suggestion regarding the need for quantitative analysis of dendritic morphology. In response, we have now included a quantitative assessment of neurite outgrowth length across experimental conditions. This has been added Figure 2E, “Neurite length,” which presents the mean neurite length (μm) for each group.
Comment 19 : Although both 2.2 m/s² and 4.0 m/s² are tested, direct statistical comparisons between these two groups are inconsistently reported. Please provide p-values or effect sizes comparing V2.2 vs. V4.0 to confirm the statistical superiority of one over the other.
Response 19 : We appreciate your insightful comments regarding the need for direct statistical comparisons between the V2.2 and V4.0 groups. To address this, we have provided comprehensive p-values for all pairwise group comparisons—including V2.2 vs. V4.0—in the newly added Supplementary Tables 1 and 2. These tables enhance the clarity of our statistical interpretation and confirm the relative effects of different acceleration conditions.
Furthermore, we have revised the main text to emphasize the statistically superior effects of 2.2 m/s² stimulation.
- (Line 528–530) "Both GAP43 and PSD95 expression levels were significantly higher in the V2.2 group compared to the V4 group (p = 0.0298 and p < 0.0001,).”
- (Line 543–544) “This difference was statistically significant (p = 0.0044), indicating the superior efficacy of the 2.2 m/s² stimulation (Supplementry Table S5.)”
- (Line 590–593) “The 2.2 m/s² condition yielded significantly greater improvements than the 4.0 m/s² condition in several key markers—including GAP43, PSD95, ChAT, and IL-1β—suggesting that moderate acceleration may be more effective for neuroprotection."
Comment 20 : In Figure 4, provide effect sizes or fold change alongside p-values for biochemical outcomes.
Response 20 : . We have revised the Results section to include both fold changes and corresponding p-values for the biochemical outcomes.
- (Line 427–428) "Vibrotactile stimulation at V2.2 significantly reduced MDA levels by 2.1-fold compared to the scopolamine group (p < 0.0001, Supplementary Table 2; Figure 4A).”
- (Line 431–434) “Vibrotactile stimulation at V2.2 and V4 restored AChE activity to levels comparable to the normal group, showing 1.3-fold reductions compared to the scopolamine group (p = 0.0084 for V2.2, p = 0.0109 for V4; Figure 4B)."
Comment 21 : It is unclear how many animals were used per group for each biochemical and histological analysis. Include exact n-values in figure legends and mention biological replicates versus technical replicates where applicable.
Response 21 : We appreciate the reviewer’s request for clarification regarding sample numbers and replicate types. In the revised manuscript, we have specified the number of animals used in each experiment within the figure legends and clearly indicated whether data represent biological or technical replicates. Behavioral experiments were conducted using biological replicates (n = 10 mice per group). For the AChE assay, cortical tissue samples from n = 4 animals per group were used as biological replicates.
However, we sincerely apologize that due to limited tissue availability, we were unable to include the same number of biological replicates for the MDA assay. As a result, MDA analysis was performed using pooled cortical tissue samples, and individual-level data from 4 animals per group could not be obtained. We will ensure more consistent sample allocation in future experiments.
Comment 22 : The results mention a “slight increase” in alternation scores without statistical significance. Avoid ambiguous language—either report precise values or state clearly that the test did not support a cognitive improvement.
Response 22 : We agree that the phrase “slight increase” may introduce ambiguity. To address this, we have revised the text to more clearly reflect the statistical outcome
- (Line 408–409) "No statistically significant differences in alternation scores were observed in the Y-maze test (Figure3E)."
Comment 23 : Figure Captions Lack Statistical Detail: Many figure legends refer to significance levels (*, **, etc.) but do not state the exact test used, the number of comparisons made, or whether corrections for multiple testing were applied.
Response 23 : We have revised all figure legends to include the specific statistical method used (Tukey’s multiple comparisons test), and clarified that the p-values represent the outcomes of post hoc comparisons following one-way ANOVA. These updates enhance the clarity and transparency of our statistical reporting in the figure legends.
Comment 24 : Behavioural Outcomes Not Linked to Molecular Changes in Real-Time: While both behaviour and molecular endpoints are assessed, they are not correlated in the results. Consider discussing whether individual-level behavioural improvements are matched with biomarker recovery.
Response 24: While direct individual-level correlation analyses were not performed, we have added discussion to better link molecular and behavioral findings.
- (Line 712–725) “play a critical role in cognitive functions. In particular, the expression of ChAT is strongly associated with the regulation of learning and memory, and its reduction has been linked to cognitive decline[110, 111]. Likewise, synaptic proteins such as PSD95, synaptophysin, and GAP43 are closely related to cortical atrophy and cognitive impairment in dementia[112, 113]. Consistent with this, our results demonstrate that the group receiving vibrotactile stimulation at 2.2 m/s² exhibited greater improvements in both cognitive performance and molecular marker expression compared to the 4.0 m/s² group. Specifically, the recognition index in the NOR test increased by 1.4-fold in the V2.2 group and 1.2-fold in the V4 group relative to the scopolamine group. In parallel, synaptic and cholinergic markers showed more prominent upregulation in the V2.2 group: PSD95 increased by 1.58-fold in V2.2 vs. 0.57-fold in V4, GAP43 by 1.44-fold vs. 1.03-fold, and ChAT by 4.89-fold vs. 3.07-fold, respectively. These findings suggest that the cognitive enhancement observed with V2.2 stimulation may be proportionally linked to the restoration of molecular targets involved in synaptic plasticity and cholinergic function.“
Discussion
Comment 25 : The discussion treats SH-SY5Y and animal model data separately. Please integrate how in vitro neuronal marker expression correlates with in vivo improvements in cognition and synaptic protein levels.
Response 25 : As suggested, we have revised the Discussion section to integrate the in vitro and in vivo findings into a unified interpretation. The following sentences have been added:
- (Line 627–631) “These cellular changes are in line with our in vivo findings, where vibrotactile stimulation improved cognitive function and increased synaptic protein expression, suggesting that the upregulation of neuronal markers such as MAP2, NeuroD1, and PSD95 at the cellular level may contribute to synaptic restoration and functional recovery in the scopolamine-induced neurotoxicity mouse model.”
- (Line 708–711) “Notably, this is consistent with our in vitro observations, where stimulation increased neuronal and synaptic markers such as MAP2, NeuroD1, and PSD95 in Aβ-treated SH-SY5Y cells, suggesting a convergent mechanism of synaptic restoration at both the cellular and in vivo levels.”
Comment 26 : The discussion should explore alternative pathways that might be modulated by vibration other than AKT/GSK3β.
Response 26 : We sincerely thank the reviewer for this valuable suggestion. In addition to the AKT/GSK3β/β-catenin pathway identified in this study, we agree that vibrotactile stimulation may also modulate alternative signaling cascades. For instance, mechanosensitive ion channels such as Piezo1 and TRP, as well as ERK/CREB signaling, have been reported to play key roles in mechanotransduction, synaptic plasticity, and neurogenesis, and may represent additional targets of vibrotactile stimulation [34–37, 123–125]. Although we did not examine these pathways in the present study, we recognize their relevance and importance. In future studies, we plan to employ transcriptomic or proteomic analyses to comprehensively characterize the broader molecular responses induced by vibrotactile stimulation, including the involvement of these alternative pathways. We appreciate the reviewer’s thoughtful recommendation, which will help us to broaden the mechanistic scope of our future work.
Comment 27 : The translational implications (e.g., for wearable vibrotactile devices or GENUS therapy) are suggested without citing clinical pilot data or discussing feasibility, dose scaling, or safety. Please temper these claims or support them with references.
Response 27 : We have revised the statements regarding clinical applicability to avoid overinterpretation. In the Discussion and Conclusion, we now clarify that our findings represent preclinical evidence and that further animal studies and clinical trials are necessary before translational application.
Comment 28 : Given that the Morris Water Maze is stress-inducing and motor-dependent, the discussion should acknowledge this limitation and its possible confounding effects on cognitive interpretation, especially with a motor-active intervention like vibration.
Response 28 : We agree with the reviewer’s comment regarding the limitations of the Morris Water Maze (MWM) in being affected by stress and motor ability. To clarify this, we have added the following sentence in the Results/Discussion section:
- (Line 641–645) "However, because MWM performance can be influenced by stress and motor ability, including potential changes induced by vibrotactile stimulation, baseline motor function was assessed prior to the experiment and two additional behavioral tests were conducted to confirm cognitive effects independently of motor performance."
Comment 29 : Since inflammatory cytokines were measured and reduced, their role in modulating synaptic plasticity (e.g., via BDNF repression or microglial pruning) should be discussed as a possible intermediary mechanism.
Response 29 : We have added a statement in the Discussion section to address this point
- (Line 690-–698) Since vibrotactile stimulation reduced inflammatory cytokines, these effects may indirectly contribute to synaptic restoration by alleviating neuroinflammation-induced suppression of synaptic plasticity pathways. In particular, pro-inflammatory cytokines such as IL-1β and TNF-α are known to downregulate BDNF expression and interfere with TrkB signaling, thereby impairing synaptic maintenance and plasticity. Additionally, chronic inflammation promotes aberrant microglial activation, which can lead to excessive synaptic pruning and loss of dendritic spines. By reducing cytokine levels and microglial activation, vibrotactile stimulation may relieve these detrimental effects and facilitate the recovery of synaptic integrity and function [99-101].
Comment 30 : The discussion does not comment on whether improved cognition was temporally or proportionally linked to the recovery of molecular markers (e.g., ChAT, PSD95). Consider correlating or at least discussing this linkage.
Response 30 : We appreciate the reviewer’s suggestion to address the relationship cognitive performance and molecular marker recovery. In response, we have added the following paragraph to the Discussion section to highlight the proportional linkage between behavioral and molecular outcomes:
- (Line 712–725) "Consistent with this, our results demonstrate that the group receiving vibrotactile stimulation at 2.2 m/s² exhibited greater improvements in both cognitive performance and molecular marker expression compared to the 4.0 m/s² group. Specifically, the recognition index in the NOR test increased by 1.4-fold in the V2.2 group and 1.2-fold in the V4 group relative to the scopolamine group. In parallel, synaptic and cholinergic markers showed more prominent upregulation in the V2.2 group: PSD95 increased by 1.58-fold in V2.2 vs. 0.57-fold in V4, GAP43 by 1.44-fold vs. 1.03-fold, and ChAT by 4.89-fold vs. 3.07-fold, respectively. These findings suggest that the cognitive enhancement observed with V2.2 stimulation may be proportionally linked to the restoration of molecular targets involved in synaptic plasticity and cholinergic function."
Comment 31 : Please compare the efficacy of vibrotactile stimulation with other non-invasive gamma entrainment methods (e.g., light, sound).
Response 31 : We sincerely appreciate the reviewer’s insightful suggestion. Comparing the efficacy of vibrotactile stimulation with other non-invasive gamma entrainment modalities such as light or auditory stimulation represents an important direction for future research. Indeed, previous studies have demonstrated that visual or auditory 40 Hz stimulation can reduce amyloid burden and modulate neuroinflammation and synaptic function in Alzheimer's disease models. While our current study focuses on the therapeutic potential of vibrotactile stimulation, we fully recognize the value of systematically evaluating its relative efficacy and mechanisms in comparison with light and sound-based approaches. We will take this constructive suggestion into account when designing future experiments to directly compare these modalities under matched conditions. Such comparative studies could provide deeper insight into the optimal stimulation parameters and delivery methods for clinical translation.

Reviewer 2 Report
Comments and Suggestions for Authors
Dear Authors
Major comment
-In general, replace the word Alzheimer in your manuscript by Amyloid beta neurotoxicity since we have not used transgenic mice of Alzheimer. We treated this human neuroblastoma cell line with Abeta olygomers. Thus, replace the word Alzheimer by Amyloid beta neurotoxicity in all the manuscript since transgenic mice are not used in your study. Thus, the conclusion should replace also this word since textually indicate ¨ This study highlights the potential of vibrotactile stimulation as a non-invasive and effective therapeutic approach for Alzheimer’s disease¨. This is not Alzheimer disease. I would recommend you change this word by Abeta neurotoxicity. Other aspect to considered is that neuroblastoma is a dopaminergic cell line although has been consistently used to study cell signalling mechanisms in Alzheimer,s disease in vitro.
-Indicate the exactly treatment for sham animals in your study
- The discussion is extense and it should be sorthened.
-Explain details for preparation of certain drugs. Scopolamine was dissolved in saline, and donepezil was dissolved in DMSO and then diluted in saline. It is necessary a deep description of these procedures. In addition, the donepezil group was administered orally donepezil (3mg/kg body weight) daily 15 minutes after scopolamine (line 127-128). Please, shall you indicate why you choose the oral route. It is not more appropriate the i.p or iv infusion of donepecile drug?
-Please, detail how MDA (Malonaldehide) levels were calculated in your TBARS assays. Since the water maze is an hippocampus dependent task and object recognition task is cortex dependent. Shall you justify the absence of determinations in the hippocampus and only in the cortex?
Additionally, indicate the calculations for standard curve interpolation for acetylcholinesterase kit. Shall you indicate the standard curve and units of Ache activity.
Please, indicate the exactly amount of proteins loaded in westerns as well as the calculations for the exactly loaded microliters by well in the gel for western blot. One example is enough.
Minnor comments
-Shall you indicate the time of exposure with DAB for immunohistochemical studies?
-Is scopolamine an inductor of Ache activity? I would expect an anticolinergic role of scopolamine as blocker of cholinergic system.
-Line 352. Most of data are convincent expect inmunohistological data of figure-6.Please, indicate the whole hippocampal area for all these inmunos of figure-6. (Figure 6. Immunohistochemical analysis of inflammatory markers IL-1β, TNF-α, and IBA1).
-line 368. the same for figure-7 (Figure 7. Immunohistochemical analysis of neuronal markers BDNF, PSD-95 expression in the hippocampus of a scopolamine-induced AD mouse model. Pl4ease, add complete detail of hippocampal formation in these figures 6 and 7.
-I have seen increases not only on IL-1 beta but also on Bax levels in scopolamine-treated rats as compare to sham animals. Does these raises a statistical significant effect. These values seems to be more than 2 x times as compare to controls in western blot analyses.
Introduction
Line 51. Please, describe how Gamma (γ) oscillations are closely associated with learning and memory processes in cytes 15 and 16.
Line 55. Explain how Sensory stimulation (GENUS) works and the possible mechanism involved in preventing oxidative and inflammatory responses against Abta42 neurotoxicity in vitro and how prevents scopolamine-induced behavioural deficits in rats.
Line 58. Particularly, explain how electromagnetic stimulation [20], auditory stimulation [21, 22], and photostimulation [23], to induce gamma oscillations and the neuroprotective contribution of each processes against Abeta-42 neurotoxicity.
Explain what factors associated to vibrotactile stimulation could explain these discrepancies in terms of neuroprotective induced-effects by vibration and the differential effect on cognitive functions in Alzheimer and taupaties.
-Shall you explain why total RNA was extracted from cells 3 days after vibrotactile stimulation at not at earlier times?
The discussion is well elaborated but it is possible to discriminate between photostimulation, sound and vibrotactile stimulation would be better.
Please, revise my suggestions
My Decision is minor revision
Thanks¡
Comments on the Quality of English LanguageThe English could be improved to more clearly express the content.
Author Response
Dear Reviewer #2
We sincerely thank for the thoughtful and detailed review of our manuscript, now titled "Acceleration-Dependent Effects of Vibrotactile Gamma Stimulation on Cognitive Recovery and Cholinergic Function in a Scopolamine-Induced Neurotoxicity Mouse Model." Your constructive feedback greatly helped us to improve the overall quality, clarity, and accuracy of the work.
In particular, following your suggestion, we have carefully replaced the term “Alzheimer’s disease” with more appropriate descriptors such as “amyloid beta neurotoxicity” or “scopolamine-induced neurotoxicity” throughout the manuscript, including the title, abstract, main text, conclusion, and figure legends..
We also addressed your comment regarding the language quality. To improve the clarity and readability of the manuscript, we had the entire text professionally edited through the MDPI Author Services. Additionally, we made significant efforts to improve the quality of all figures by maximizing image resolution and enhancing contrast where needed, particularly for Western blot and immunohistochemistry images.
Further, we provided more detailed methodological descriptions, including drug preparation procedures, sham treatment conditions, quantification methods for MDA and AChE, and the amount of protein loaded in Western blots. We also clarified the rationale for the use of SH-SY5Y cells and acknowledged their dopaminergic characteristics in the Discussion. In response to your suggestions, we added hippocampal-wide images for Figures 6 and 7 as supplementary figures, improved explanation of GENUS mechanisms across different modalities (light, sound, vibration), and distinguished their respective neuroprotective contributions. The Discussion section was revised for better organization and shortened to enhance clarity while maintaining the scientific depth.
We have addressed all of your comments carefully in the point-by-point response below. We truly appreciate your valuable feedback, which has greatly improved the manuscript, and we hope that the revised version meets your expectations.
Sincerely,
Tae-Woo Kim
Department of Biomedical Engineering, Dongguk University, Goyang-si 10326, Republic of Korea
Email: xodn8876@naver.com
Tel: +82-10-5513-8876
Corresponding Author: Prof. Young-Kwon Seo
Email: bioseo@dongguk.edu
Tel: +82-10-8502-9916
Major comment
Comments 1: In general, replace the word Alzheimer in your manuscript by Amyloid beta neurotoxicity since we have not used transgenic mice of Alzheimer. We treated this human neuroblastoma cell line with Abeta olygomers. Thus, replace the word Alzheimer by Amyloid beta neurotoxicity in all the manuscript since transgenic mice are not used in your study. Thus, the conclusion should replace also this word since textually indicate ¨ This study highlights the potential of vibrotactile stimulation as a non-invasive and effective therapeutic approach for Alzheimer’s disease¨. This is not Alzheimer disease. I would recommend you change this word by Abeta neurotoxicity.
Response 1 : Thank you for this important comment. We fully agree with your suggestion.
Therefore, we have revised the entire manuscript to replace “Alzheimer” with “neurotoxicity” or “scopolamine-induced neurotoxicity model” as appropriate. This change has been applied consistently throughout the manuscript, including the title, abstract, main text, figure legends, and conclusion.
Comments 2: Other aspect to considered is that neuroblastoma is a dopaminergic cell line although has been consistently used to study cell signalling mechanisms in Alzheimer,s disease in vitro.
Response 2 : Indeed, SH-SY5Y cells in their undifferentiated state exhibit dopaminergic characteristics. However, as noted, this cell line has been widely adopted for in vitro studies on Alzheimer’s disease mechanisms due to its responsiveness to Aβ toxicity and ease of neuronal differentiation. To partially address this limitation, we used retinoic acid (RA) to induce neuronal differentiation, which promotes cholinergic features. Nonetheless, we agree that RA-induced differentiation does not fully replicate the phenotype of mature cholinergic neurons, and we have now acknowledged this limitation in the Discussion section:
- (Line 615–620) "Although SH-SY5Y cells possess dopaminergic characteristics in their undifferentiated state, RA-induced differentiation can promote cholinergic features. However, RA treatment alone does not fully recapitulate the phenotype of mature cholinergic neurons. Therefore, the current findings should be interpreted with caution, and future studies employing fully differentiated cholinergic neuron models may provide more physiologically relevant insights."
Comments 3 : Indicate the exactly treatment for sham animals in your study
Response 3 : We thank the reviewer for the helpful comment. We have clarified the treatment of the sham group in the Methods section. Specifically, we have added the following sentence:
- (Line 222–224) "Sham group received an intraperitoneal injection of saline (3 mg/kg body weight) and were placed on the vibration platform without stimulation for 30 minutes per day."
Comments 4 : The discussion is extense and it should be sorthened.
Response 4 : Thank you very much for your valuable comment. In response to your suggestion, we thoroughly reviewed the Discussion section with the aim of enhancing clarity and conciseness. Redundant descriptions and overlapping statements were carefully rephrased or removed to avoid unnecessary repetition. Wherever possible, lengthy explanations were condensed into more succinct and focused paragraphs without compromising the scientific content or integrity of the discussion. As a result of these efforts, we reduced the overall word count from approximately 2500 words to around 1800 words.
Comments 5 : Explain details for preparation of certain drugs. Scopolamine was dissolved in saline, and donepezil was dissolved in DMSO and then diluted in saline. It is necessary a deep description of these procedures.
Response 5 : Thank you for your helpful suggestion. In response, we have revised the Methods section to describe the drug preparation process in greater detail.
- (Line 224–228) “Scopolamine hydrobromide was dissolved in saline at a concentration of 0.3 mg/mL and administered intraperitoneally at a volume of 10 µL/g. Donepezil (10 mg) was dissolved in 100 µL DMSO and diluted with saline to 33 mL, resulting in a 0.3 mg/mL solution with 0.3% DMSO (v/v). Donepezil was administered orally at 10 µL/g, 15 minutes after scopolamine injection.”
Comments 6 : In addition, the donepezil group was administered orally donepezil (3mg/kg body weight) daily 15 minutes after scopolamine (line 127-128). Please, shall you indicate why you choose the oral route. It is not more appropriate the i.p or iv infusion of donepecile drug?
Response 6 : In this study, we selected the oral route because it is the traditional and clinically relevant method for Donepezil delivery. This approach aligns with several previous studies that administered Donepezil orally in mouse models of neurodegeneration [46, 49, 50]. Additionally, we aimed to minimize potential interactions with scopolamine, which was administered via intraperitoneal injection. Using the same i.p. route for both drugs could lead to unforeseen pharmacodynamic or pharmacokinetic interactions. Therefore, to ensure clearer interpretation of treatment effects and to avoid potential route-related confounding variables, we opted for oral administration of Donepezil.
“The oral route of administration was selected based on previous studies using mouse models [46, 49, 50]. Additionally, we aimed to minimize potential interactions with scopolamine, which was administered via intraperitoneal injection.”
Comment 7 : Please, detail how MDA (Malonaldehide) levels were calculated in your TBARS assays.
Response 7 : To address the first part of the comment, we have now included detailed information on how MDA levels were calculated in the Methods section. And the standard curve is now provided in the Supplementary (Supplementary Figure S1).
- (Line 304–311) “The optical density (OD) of each sample and standard was measured at 543 nm, and the OD of the blank was subtracted. Duplicate readings were averaged, and MDA concentrations were calculated using a standard curve generated from known MDA concentrations. The standard curve was generated by performing a 2-fold serial dilution starting from 400 μM, and the resulting graph is presented in the Supplementary Information (Supplementary Figure S1). Final values were adjusted for any dilution factor and normalized to total protein concentration, expressed as μM/mg protein.”
Comment 8 : Since the water maze is an hippocampus dependent task and object recognition task is cortex dependent. Shall you justify the absence of determinations in the hippocampus and only in the cortex?
Response 8 : While it is well-established that the Morris Water Maze primarily reflects hippocampus-dependent learning, and the object recognition task reflects cortex-dependent memory, our decision to focus on cortical tissue was based on both practical and experimental grounds. Specifically, our previous work, “Comparison of Malondialdehyde, Acetylcholinesterase, and Apoptosis-Related Markers in the Cortex and Hippocampus of Cognitively Dysfunctional Mice Induced by Scopolamine” (Hee-Jung Park, 2024), demonstrated that scopolamine-induced increases in MDA levels were more pronounced in the cortex than in the hippocampus. Due to limited tissue availability, hippocampal samples were reserved for histological and Western blot analyses, while cortical tissue was used for biochemical assays (MDA and AChE). Nonetheless, the absence of hippocampal biochemical data is a limitation, and future studies should analyze both regions for comprehensive insight.
Comment 9 : Additionally, indicate the calculations for standard curve interpolation for acetylcholinesterase kit. Shall you indicate the standard curve and units of Ache activity.
Response 9 : To address the comment, we have now included detailed information on how AChE levels were calculated in the Methods section. And the standard curve is now provided in the Supplementary (Supplementary Figure S1).
- (Line 316–325) “According to the manufacturer’s instructions, cortical tissue was homogenized in PBS, and 50 µL of sample or standard was mixed with 50 µL of working solution containing DTNB and acetylthiocholine. After incubation at room temperature for 30 minutes, absorbance was measured at 410 nm. A standard curve was generated by performing 1:3 serial dilutions starting from a 1000 mU/mL acetylcholinesterase standard solution. The resulting values were analyzed using an online linear regression calculator, and the curve was plotted on a semi-logarithmic scale (Supplementary Figure S1).The activity in samples was calculated by interpolation using the linear regression equation derived from the standard curve.”
Comment 10 : Please, indicate the exactly amount of proteins loaded in westerns as well as the calculations for the exactly loaded microliters by well in the gel for western blot. One example is enough.
Response 10 : As requested, we have clarified the protein loading details for the Western blot experiments.
- (Line 330–336) “Protein concentrations were measured using a BCA assay by generating a standard curve with bovine serum albumin (BSA) and calculating the protein concentration of samples based on their absorbance values. For SDS-PAGE, the sample was loaded onto a 10% SDS-polyacrylamide gel and electrophoresed at 90 V for 120 min. The average protein concentration was approximately 6 µg/µL. To load 30 µg of protein per lane, 4.9 µL of lysate was mixed with loading buffer to a final volume of 20 µL per well.”
Minnor comments
Comment 11 : Shall you indicate the time of exposure with DAB for immunohistochemical studies?
Response 11 : As requested, we have specified the DAB exposure time in the immunohistochemistry protocol.
- (Line 335–357) “Color development was performed using DAB (Agilent Dako, #K5007) for 3 minutes for all markers, except for BDNF and PSD95, which required 4 minutes to achieve optimal signal.”
Comment 12 : Is scopolamine an inductor of Ache activity? I would expect an anticolinergic role of scopolamine as blocker of cholinergic system.
Response 12 : While scopolamine is indeed a muscarinic acetylcholine receptor antagonist that exerts anticholinergic effects by blocking cholinergic signaling, several studies have reported that scopolamine administration can paradoxically lead to increased acetylcholinesterase (AChE) activity in the brain. This upregulation is reflect a compensatory mechanism or a secondary response to impaired cholinergic neurotransmission. Accordingly, the observed increase in AChE activity in our study is consistent with previous reports documenting enhanced AChE activity following scopolamine treatment [52, 94, 95].
Comment 13 : Most of data are convincent expect inmunohistological data of figure-6.Please, indicate the whole hippocampal area for all these inmunos of figure-6. (Figure 6. Immunohistochemical analysis of inflammatory markers IL-1β, TNF-α, and IBA1).
Response 13 : As requested, we have included a supplementary figure (Supplementary Figure S3) showing the full hippocampal region for IL-1β, TNF-α, and IBA1 staining across all experimental groups. These 40× magnification images better illustrate the overall distribution of inflammatory markers throughout the hippocampus.
Comment 14 :. the same for figure-7 (Figure 7. Immunohistochemical analysis of neuronal markers BDNF, PSD-95 expression in the hippocampus of a scopolamine-induced AD mouse model. Pl4ease, add complete detail of hippocampal formation in these figures 6 and 7.
Response 14 : As requested, we have included supplementary images (Supplementary Figure S4) showing the entire hippocampal formation stained for BDNF and PSD95 to support the data presented in Figure 7.
Comment 15 : I have seen increases not only on IL-1 beta but also on Bax levels in scopolamine-treated rats as compare to sham animals. Does these raises a statistical significant effect. These values seems to be more than 2 x times as compare to controls in western blot analyses.
Response 15 : We thank the reviewer for this valuable comment. To clarify, we have now included detailed statistical analyses for both IL-1β and BAX expression levels. If the reviewer found the band-to-graph correspondence unclear, we have re-presented the β-actin loading control with enhanced contrast in the revised figure to ensure more distinct normalization and visualization.
- (Line 504–513)"The expression of IL-1β was significantly increased in the scopolamine group compared to the sham group, showing a 4.18-fold increase (p < 0.0001). However, both the Donepezil and vibrotactile stimulation (V2.2, V4) groups exhibited a significant reduction in IL-1β levels (Figure 8B; Supplementary Table 2). Similarly, BAX was markedly upregulated in the scopolamine group, showing a 2.84-fold increase compared to the sham group (p = 0.0002), indicating enhanced neuronal apoptosis. Notably, the V2.2 group demonstrated a greater reduction in BAX expression than the other treatment groups, with a 2.98-fold decrease compared to the scopolamine group (p = 0.0001; Figure 8C)."
Introduction
Comment 16 : Line 51. Please, describe how Gamma (γ) oscillations are closely associated with learning and memory processes in cytes 15 and 16.
Response 16 : we have added a description of the relationship between gamma oscillations and learning/memory with the appropriate references, as follows:
- (Line 59–63) “Gamma oscillations strengthen synaptic plasticity and promote efficient information transfer across brain regions. They also organize the temporal structure of memory traces, supporting both memory formation and retrieval. Furthermore, by enhancing synchronization between the prefrontal cortex and hippocampus, gamma activity contributes to higher-order cognitive functions such as learning, attention, and working memory [22,23].”
Comment 17 : Line 55. Explain how Sensory stimulation (GENUS) works and the possible mechanism involved in preventing oxidative and inflammatory responses against Abta42 neurotoxicity in vitro and how prevents scopolamine-induced behavioural deficits in rats.
Response 17 : We added a paragraph describing the proposed mechanisms of sensory stimulation (GENUS) based on recent studies. The following sentences were inserted:
- (Line 93–102) “Although GENUS has shown promising neuroprotective effects, the precise mechanisms by which it influences amyloid β deposition remain incompletely understood. Some studies have reported that gamma entrainment or GENUS promotes a phagocytic phenotype in microglia, thereby enhancing their clearance activity, while others have demonstrated that 40 Hz stimulation reduces amyloid burden through a combination of decreased APP cleavage intermediates in hippocampal neurons and increased microglial endocytosis [29, 30, 38]. These findings suggest that different sensory stimulation modalities can enhance gamma oscillations through complex neural circuits; however, the cellular and molecular processes by which such gamma synchronization leads to memory improvement remain largely unresolved and represent an important topic for future investigation.”
Comment 18 : Line 58. Particularly, explain how electromagnetic stimulation [20], auditory stimulation [21, 22], and photostimulation [23], to induce gamma oscillations and the neuroprotective contribution of each processes against Abeta-42 neurotoxicity.
Response 18 : We have added an explanation describing how these different types of stimulation induce gamma oscillations and provide neuroprotection, as follows:
- (Line 603–615) “When 40 Hz stimulation is repeatedly delivered, the sensory input entrains the corresponding sensory cortex, aligning brain oscillations to 40 Hz. This amplifies the brain’s natural gamma rhythms, which underlies the mechanism of GENUS. Photostimulation have been shown to effectively entrain gamma oscillations across widespread brain regions, reducing neuronal and synaptic loss and thereby ameliorating neuropathology. These approaches also attenuated inflammation and significantly reduced amyloid levels in the visual cortex, contributing to a decrease in plaque pathology [28,60,61]. Non-contact acoustic stimulation entrains 40 Hz gamma activity in the auditory cortex and hippocampus, leading to reductions in Aβ deposition, inhibition of GSK 3β, recovery of mitochondrial function, and improvements in cognitive performance [28,60]. Electromagnetic stimulation has been shown to induce gamma oscillations by modulating cortical excitability, enhancing cholinergic function, and restoring abnormal connectivity between the hippocampus and prefrontal cortex [24].”
Comment 19 : Explain what factors associated to vibrotactile stimulation could explain these discrepancies in terms of neuroprotective induced-effects by vibration and the differential effect on cognitive functions in Alzheimer and taupaties.
Response 19 : We have revised the Discussion section to address how specific parameters of vibrotactile stimulation (e.g., acceleration, frequency, and duration) may influence the neuroprotective outcomes observed in different disease models. The following sentence has been newly added:
- (Line 742–753) " The neuroprotective effects of vibrotactile stimulation are strongly influenced by specific stimulation parameters, such as acceleration, frequency, and duration. Importantly, these parameters do not act independently but rather in a synergistic and interdependent manner. Their combined effects determine the activation threshold and dynamics of mechanosensory receptors, including Piezo1 and TRP channels, which, in turn, modulate downstream signaling pathways such as AKT/GSK3β and ERK/CREB [32,114-119]. This complex interplay can influence a wide range of biological processes, including neuroinflammation, synaptic plasticity, and cellular resilience. As such, both the molecular mechanisms and functional outcomes may vary depending on the specific combination of stimulation parameters. Consequently, even subtle changes in stimulation conditions may lead to markedly different neuroprotective and cognitive effects, potentially explaining the variability observed across studies and disease models [32,120]"
Comment 20 : Shall you explain why total RNA was extracted from cells 3 days after vibrotactile stimulation at not at earlier times?
Response 20 : We apologize for the confusion. Total RNA was extracted 30 minutes after the final vibrotactile stimulation session, which was performed over a 3-day period. This timing was chosen to capture the immediate gene expression changes following the final stimulation. The method description has been corrected to reflect this accurately.
- (Line 174–175)“Total RNA was extracted from cells 30 minutes after the final day of vibrotactile stimulation using RNAiso Plus (1 mL per sample; Takara).”
Comment 21 : The discussion is well elaborated but it is possible to discriminate between photostimulation, sound and vibrotactile stimulation would be better.
Response 21 : We appreciate the reviewer’s valuable comment. In the revised Discussion section, we have carefully clarified the distinctions between photostimulation, sound-based stimulation, and vibrotactile stimulation. Efforts were made to explicitly describe the mechanisms and effects of each modality where relevant, and to present them in a more organized and coherent manner to enhance clarity and readability.

Round 2
Reviewer 1 Report
Comments and Suggestions for Authors
I commend the authors for their significant improvements, which have enhanced the clarity, methodological rigour, and integration of findings in the manuscript. The revised Introduction now offers a more robust justification for acceleration values, a molecular overview of the AKT/GSK3β/β-catenin pathway, and a discussion of clinical translation challenges. However, several comments need to be addressed before considering this manuscript for publication.
COMMENTS:
1] While the primer sequences have been provided, it is concerning that each experimental condition was assessed using RNA from only one biological sample per group, even with three technical replicates. Technical replicates mitigate assay variability but fail to account for biological variability, which is crucial for dependable and generalizable results.
It is advised to conduct a minimum of three separate biological replicates for RT-qPCR to guarantee statistical robustness and reproducibility.
2] For the MDA assay, pooled cortical tissue was used, which precludes assessment of inter-animal variability. This should be discussed as a limitation, particularly given that oxidative stress measures are often variable across individuals.
Author Response
Dear Reviewer #1,
We sincerely thank you for your constructive and encouraging feedback on our revised manuscript, now titled "Acceleration-Dependent Effects of Vibrotactile Gamma Stimulation on Cognitive Recovery and Cholinergic Function in a Scopolamine-Induced Neurotoxicity Mouse Model." We are grateful that you recognized the significant improvements in clarity, methodological rigor, and integration of findings from our previous revision. Your remaining comments regarding biological replicates for RT-qPCR and the pooling strategy in the MDA assay were highly valuable and have guided further refinement of our work.
In response, we have:
- Clarified in the Results and Discussion sections that SH-SY5Y RT-qPCR experiments were conducted in three independent biological replicates, but only one representative dataset was presented due to variability in error bars. We have explicitly acknowledged this as a limitation and emphasized the importance of integrating multiple biological replicates in future studies to enhance statistical robustness.
- Addressed the limitation of pooling cortical tissues for the MDA assay by adding a clear statement in the Discussion that this approach precludes assessment of inter-animal variability. We have also noted that oxidative stress markers can vary considerably between individuals and suggested that future work should incorporate individual-level analyses for more reliable and generalizable conclusions.
These changes reflect our sincere effort to transparently acknowledge methodological constraints while maintaining the scientific integrity of the study. We have carefully revised the relevant sections to ensure that these points are clearly communicated.
We kindly invite you to review our detailed point-by-point responses below. We greatly appreciate your time, expertise, and continued guidance, and we hope that the revised manuscript now fully meets your expectations.
Sincerely,
Tae-Woo Kim
Department of Biomedical Engineering, Dongguk University, Goyang-si 10326, Republic of Korea
Email: xodn8876@naver.com
Tel: +82-10-5513-8876
Corresponding Author: Prof. Young-Kwon Seo
Email: bioseo@dongguk.edu
Tel: +82-10-8502-9916
COMMENTS:
Comment 1 : While the primer sequences have been provided, it is concerning that each experimental condition was assessed using RNA from only one biological sample per group, even with three technical replicates. Technical replicates mitigate assay variability but fail to account for biological variability, which is crucial for dependable and generalizable results. It is advised to conduct a minimum of three separate biological replicates for RT-qPCR to guarantee statistical robustness and reproducibility. We agree that presenting only one dataset may limit statistical robustness and generalizability.
Response 1 : We thank the reviewer for this valuable comment. In our RT-qPCR experiments using SH-SY5Y cells, we performed three independent biological replicates in total. All three experiments showed a similar trend in gene expression changes; however, we presented the dataset with the most distinct results in the figures to clearly illustrate the effects. Although combining all datasets would have increased the sample size, the resulting error bars became excessively large due to inter-experimental variability, which could obscure the observed trends. We agree that presenting only one dataset limits the statistical robustness and generalizability of the results. The following sentences were added:
- (Line 387–389) “These experiments were repeated three times under identical conditions, and similar trends were observed in all trials. Therefore, only one representative dataset was presented in the figures.”
- (Line 636–639) “Only one of the three repeated experiments was included in the presentation of the data. We acknowledge that this approach may reduce statistical robustness, and recommend that future studies integrate multiple biological replicates to improve the reliability and generalizability of the findings.”
Comment 2 : For the MDA assay, pooled cortical tissue was used, which precludes assessment of inter-animal variability. This should be discussed as a limitation, particularly given that oxidative stress measures are often variable across individuals.
Response 2 : We appreciate the reviewer’s insightful comment. In the present study, cortical tissues from multiple animals within each group were pooled and reduced to two data points for the MDA assay. This approach was chosen to obtain an overall average value, but it inevitably prevents direct evaluation of inter-animal variability. We acknowledge that oxidative stress markers can vary considerably among individuals, and thus this pooling strategy is a limitation of the current study.
- (Line 429–431) “The cortical tissues from multiple animals in each group were pooled and reduced to two data points, which were presented as average values.”
- (Line 429–431) “In the present study, cortical tissues from multiple animals were pooled and reduced to two data points. This approach does not allow for direct assessment of inter-animal variability, which is particularly relevant given that oxidative stress markers can vary considerably between individuals. While we sought to represent the results using average values, future studies should perform individual analyses to provide more reliable and generalizable conclusions.”

Round 3
Reviewer 1 Report
Comments and Suggestions for Authors
While I appreciate the authors’ efforts to address the previous round of comments, that has improved the overall quality and clarity of the manuscript. However, the following points require further attention and incorporation:
Comment 1: The authors' explanation of the RT-qPCR experiments is helpful, but publishing only one dataset instead of aggregated data is unscientific. When conducting quantitative PCR tests on cell lines like SH-SY5Y cells, it is recommended to publish the mean ± standard deviation (or standard error) for all biological replicates, regardless of variability. Only showing the “most distinct” dataset adds selection bias and reduces the statistical validity of the results by not reflecting biological system diversity.
I strongly advise authors to:
A] Aggregate all biological replicates (n=3) and present the mean ± SD or SEM.
B] Include individual data points in the figure (e.g., scatter plots overlaid on bar graphs) to visualise variability.
Comment 2: The identical β-actin blot seems to have been utilized as a loading control for several target proteins, as in Figures 8 and 9. Each target protein must have a comparable loading control from the same gel or membrane. Utilizing the identical β-actin picture across many investigations without proper elucidation presents issues pertaining to data integrity and repeatability. The authors must supply the original, complete, uncropped blots for each protein to verify appropriate experimental procedures.

Author Response
Dear Reviewer #1,
We sincerely thank you for your constructive and encouraging feedback on our manuscript, now titled "Acceleration-Dependent Effects of Vibrotactile Gamma Stimulation on Cognitive Recovery and Cholinergic Function in a Scopolamine-Induced Neurotoxicity Mouse Model." We greatly appreciate your recognition of the improvements in clarity, methodological detail, and overall presentation achieved through the previous revisions.
Your remaining comments regarding the presentation of biological replicates in the RT-qPCR experiments and the use of identical β-actin loading controls in multiple Western blot panels were highly valuable and have guided further refinement of our work.
In response, we have:
- RT-qPCR – Aggregated all three independent biological replicates (n = 3) for each condition, with each replicate representing the mean of three technical replicates. Data are presented as mean ± SEM with individual points. The control group, normalized to 1 by the 2^-ΔΔCt method, shows no visible error bars. Supplementary RT-qPCR figures were also updated for consistency.
- Western blot – Clarified in the Methods and figure legends that all blots used identical protein extracts from the same sample set. A single β-actin blot was used as a representative loading control due to identical loading conditions, and all uncropped blots are now provided in Supplementary Figure S5. While separate loading controls would enhance clarity, repeating experiments was not feasible; we will apply this approach in future studies.
These revisions reflect our commitment to transparency, methodological rigor, and addressing the reviewer’s concerns directly, while maintaining the scientific integrity of the study. We have carefully revised the relevant sections to ensure these points are clearly communicated.
We kindly invite you to review our detailed point-by-point responses below. We deeply appreciate your time, expertise, and continued guidance, and we hope that the revised manuscript now fully meets your expectations.
Sincerely,
Tae-Woo Kim
Department of Biomedical Engineering, Dongguk University
Goyang-si 10326, Republic of Korea
Email: xodn8876@naver.com
Tel: +82-10-5513-8876
Corresponding Author: Prof. Young-Kwon Seo
Email: bioseo@dongguk.edu
Tel: +82-10-8502-9916
Comment 1: The authors' explanation of the RT-qPCR experiments is helpful, but publishing only one dataset instead of aggregated data is unscientific. When conducting quantitative PCR tests on cell lines like SH-SY5Y cells, it is recommended to publish the mean ± standard deviation (or standard error) for all biological replicates, regardless of variability. Only showing the “most distinct” dataset adds selection bias and reduces the statistical validity of the results by not reflecting biological system diversity. I strongly advise authors to:
A] Aggregate all biological replicates (n=3) and present the mean ± SD or SEM.
B] Include individual data points in the figure (e.g., scatter plots overlaid on bar graphs) to visualize variability.
Response 1 : We sincerely appreciate the reviewer’s constructive suggestion. In the revised manuscript, we have re-analyzed the RT-qPCR data to aggregate all three independent biological replicates (n=3) for each experimental condition. Each biological replicate consisted of three technical replicates, and the mean value of these technical replicates was calculated and used as a single data point for that biological replicate. The results are now presented as the mean ± standard error of the mean (SEM) from the three biological replicates, with individual data points shown in the figures to illustrate variability.
In RT-qPCR analysis, all values were calculated as relative expression levels using the 2^-ΔΔCt method, where the control group is designated as the calibrator with a mean value of 1. Due to this normalization process, all biological replicates of the control group are fixed at 1, regardless of inherent variability, and thus no error bars are visible for this group.
In addition, the Supplementary Figures containing RT-qPCR data have also been updated to use the same format, ensuring consistency across the main and supplementary materials.
- (Line 193–198 ) “Each experimental condition was analyzed in three independent biological replicates, and within each biological replicate, three technical replicates were performed. For data analysis, the mean value of the three technical replicates was calculated for each biological replicate. These averaged values were then used to compute the mean ± standard error of the mean (SEM) across the three biological replicates, and individual data points in the figures represent each biological replicate’s averaged value.”
- (Figure 2 legends line 398–399) “As the control group (aβ-) values were normalized to 1, all biological replicates showed identical values, and therefore no error bars are visible for this group”
Comment 2: The identical β-actin blot seems to have been utilized as a loading control for several target proteins like in Figure 8 and 9. Each target protein must have a comparable loading control from the same gel or membrane. Utilizing the identical β-actin picture across many investigations without proper elucidation presents issues pertaining to data integrity and repeatability. The authors must supply the original, complete, uncropped blots for each protein to verify appropriate experimental procedures.
Response 2 : We thank the reviewer for the valuable comment. All western blot experiments in Figures 8 and 9 were performed using identical protein extracts from the same set of samples, loaded in equal amounts onto separate gels for probing with different target antibodies. A single β-actin blot was obtained from one of these gels as a representative loading control, since the loading conditions and samples were identical across all experiments. For transparency, we have provided the original, complete, uncropped blots for all western blot images in Supplementary S5. In addition, we have explicitly stated in the Methods section that all western blot experiments were performed using the same sample set, and indicated in the figure legends of Figures 8 and 9 that the same β-actin image was used as a representative loading control.
We acknowledge that using a separate loading control for each target blot would further enhance data clarity. However, because the experiments were conducted on a limited set of precious samples that are no longer available, repeating the blots to generate additional β-actin images is unfortunately not feasible. We will adopt the approach of including a matched loading control for every blot in future experiments to avoid any potential confusion.
- (Line 340–342) “All western blot experiments were performed using identical protein extracts from the same set of samples, which were loaded in equal amounts for each target protein on separate gels.”
- (Line 350–352)“A single β-actin blot obtained from one of these gels was used as a representative loading control for all target proteins because the loading conditions and samples were identical across all experiments.”
